# Associations between alcohol consumption and gray and white matter volumes in the UK Biobank

Remi Daviet[1]✉, Gökhan Aydogan[2], Kanchana Jagannathan[3], Nathaniel Spilka[3], Philipp D. Koellinger [4,5], Henry R. Kranzler [3,6], Gideon Nave [7]✉ & Reagan R. Wetherill [3]✉

Heavy alcohol consumption has been associated with brain atrophy, neuronal loss, and poorer white matter fiber integrity. However, there is conflicting evidence on whether light-to-moderate alcohol consumption shows similar negative associations with brain structure. To address this, we examine the associations between alcohol intake and brain structure using multimodal imaging data from 36,678 generally healthy middle-aged and older adults from the UK Biobank, controlling for numerous potential confounds. Consistent with prior literature, we find negative associations between alcohol intake and brain macrostructure and microstructure. Specifically, alcohol intake is negatively associated with global brain volume measures, regional gray matter volumes, and white matter microstructure. Here, we show that the negative associations between alcohol intake and brain macrostructure and micro-structure are already apparent in individuals consuming an average of only one to two daily alcohol units, and become stronger as alcohol intake increases.

[1] Wisconsin School of Business, University of Wisconsin–Madison, Madison, WI, USA. [2] Department of Economics, University of Zurich, Zurich, Switzerland. [3] Department of Psychiatry, Perelman School of Medicine, University of Pennsylvania, Philadelphia, PA, USA. [4] La Follette School of Public Affairs, University of Wisconsin-Madison, Madison, WI, USA. [5] Department of Economics, School of Business and Economics, Vrije Universiteit Amsterdam, Amsterdam, the Netherlands. [6] Crescenz VAMC, Philadelphia, PA, USA. [7] Marketing Department, The Wharton School, University of Pennsylvania, Philadelphia, PA, USA. ✉email: daviet@wisc.edu; gnave@wharton.upenn.edu; rweth@pennmedicine.upenn.edu

Alcohol consumption is one of the leading contributors to the global burden of disease and to high healthcare and economic costs. Alcohol use disorder (AUD)[1] is one of the most prevalent mental health conditions worldwide[2], with harmful effects on physical, cognitive, and social functioning[3]. Chronic excessive alcohol consumption is associated with direct and indirect adverse effects, including (but not limited to) cardiovascular disease[4], nutritional deficiency[5], cancer[6], and accelerated aging[7–9].

Chronic alcohol use is associated with changes in brain structure and connectivity[9–11]. Neuroimaging studies have shown that chronic heavy alcohol consumption (3 or more drinks for women and 4 or more drinks for men on any day) is associated with widespread patterns of macrostructural and microstructural changes, primarily affecting frontal, diencephalic, hippocampal, and cerebellar structures[9,10,12]. A recent meta-analysis of individuals with AUD ($n = 433$) showed lower gray matter volume (GMV) in the corticostriatal-limbic circuits, including regions of the prefrontal cortex, insula, superior temporal gyrus, striatum, thalamus, and hippocampus compared to healthy controls ($n = 498$)[13]. Notably, lower GMV in striatal, frontal, and thalamic regions was associated with AUD duration or lifetime alcohol consumption. Although alcohol consumption can produce global and regional tissue volume changes, frontal regions are particularly associated with these effects[14–16]. Further, research suggests that the effects of alcohol consumption on brain volume interact with the effects of aging[9,17].

Alcohol-consumption related white matter (WM) microstructural alterations are a hallmark change associated with AUD[18–20]. Neuroimaging studies have consistently shown WM degeneration of the corpus callosum in AUD[3,21,22]. However, the effects of AUD on WM microstructure, as evidenced by decreased fractional anisotropy (FA) and increased mean diffusivity (MD), are not limited to the corpus callosum but are also seen in the internal and external capsules, fornix, frontal forceps, superior cingulate, and longitudinal fasciculi[3,21,23]. Further, research indicates that anterior and superior WM systems are more likely to show changes associated with AUD than posterior and inferior systems[24]. However, because conventional diffusion tensor imaging (DTI) measures (FA and MD) are based on a simplistic brain tissue microstructure model, they fail to account for the complexities of neurite geometry[25]. For example, the lower FA observed in individuals with AUD may reflect lower neurite density and/or greater orientation dispersion of neurites, which conventional DTI measures do not differentiate[26,27].

Despite an extensive literature on the associations of alcohol consumption with brain structure and microstructure in individuals with AUD, there is limited research exploring these associations in individuals who consume alcohol but do not have AUD. In some studies of middle-aged and older adults, moderate alcohol consumption was associated with lower total cerebral volume[28], gray matter atrophy[29,30], and lower density of gray matter in frontal and parietal brain regions[30]. However, other studies have failed to show an association[31], and one study showed a positive association of light-to-moderate alcohol consumption and GMV in older men[32]. One interpretation of these findings is that an inverse U-shaped, dose-dependent association exists between alcohol consumption and brain structure[32]. However, this interpretation was not supported by a longitudinal cohort study, which showed no difference in structural brain measures between individuals who did not consume alcohol and those who consumed between 1 and <7 alcohol units per week (i.e., light alcohol consumption), while individuals who consumed moderate-to-high amounts of alcohol (i.e., 14 or more alcohol units per week) showed GMV atrophy in the hippocampi and altered WM microstructure (lower FA, higher MD) in the corpus callosum[33].

The inconclusive evidence regarding the association between moderate alcohol intake and brain structure in the general population may be because the literature consists of mostly small, unrepresentative studies with limited statistical power[34,35]. Moreover, most studies have not accounted for the effects of many relevant covariates and, therefore, have yielded potentially biased findings. Potential confounds that may be associated with individual differences in both alcohol intake and neuroanatomy include sex[36], body mass index (BMI)[37], age[38,39], and genetic population structure[40]. Similar to other research fields, progress in this area may also be limited by publication bias, as positive findings are more likely to be published than null results[41].

The current study examines the associations between alcohol intake and measures of GM structure and WM microstructure in the brain in a large population sample. We perform a pre-registered analysis of multimodal imaging data from the UK Biobank (UKB)[42–44], which controls for numerous potential confounds. The UKB, a prospective cohort study representative of the United Kingdom (UK) population aged 40–69 years, is the largest available collection of high-quality MRI brain scans, alcohol-related behavioral phenotypes, and measurements of the socioeconomic environment. The UKB brain imaging data include three structural modalities, resting and task-based fMRI, and diffusion imaging[42–44]. The WM fiber integrity measures available in the UKB include the conventional FA and MD metrics and neurite orientation dispersion and density imaging (NODDI) measures[26]. Such measures offer information on WM microstructure and estimates of neurite density (i.e., intracellular volume fraction; ICVF), extracellular water diffusion (i.e., isotropic volume fraction; ISOVF), and tract complexity/fanning (i.e., orientation dispersion, OD). Specifically, we assess associations between alcohol intake (i.e., mean daily alcohol units; one unit = 10 ml or 8 g of ethanol) and imaging derived phenotypes (IDPs) of brain structure (total GMV, total WMV, and 139 regional GMVs), as well as 375 IDPs of WM microstructure (DTI and NODDI indices), using data from 36,678 UKB participants. Our analyses adjust for numerous covariates (see Methods for full list).

Our sample size provides us statistical power of 90% to detect effects as small as $f^2 < 0.00078$ at the 5% significance level, after accounting for multiple hypotheses testing ($p_{uncorrected} < 1.64 \times 10^{-4}$). Given previous findings, we hypothesized a negative relationship between alcohol intake and global GMV and WMV in individuals who consume large amounts of alcohol (i.e., females who report consuming more than 18 units/week and males who report consuming more than 24 units/week). The large general population sample provided sufficient sensitivity to qualitatively and quantitatively assess how associations vary across the drinking spectrum and test at which threshold associations emerge. Our design also allowed us to explore whether the associations between alcohol intake and GMV and WM microstructure are localized in specific regions or widespread across the brain and compare the associations across various WM integrity indicators.

## Results
Figure 1 summarizes the characteristics of the 36,678 participants (52.8% female), including the distributions of age, daily alcohol units, and global GMV and WMV. Participants were healthy adults of middle age or older. We normalize all IDPs for head size by multiplying the raw IDP by the head size scaling factor.

**Relationship between Global GMV and WMV and alcohol intake**. The scatter plots in Figs. 2 and 3 illustrate the relationships of the global IDPs (GMV and WMV, both normalized for head size) with age and daily alcohol intake (in log scale) within

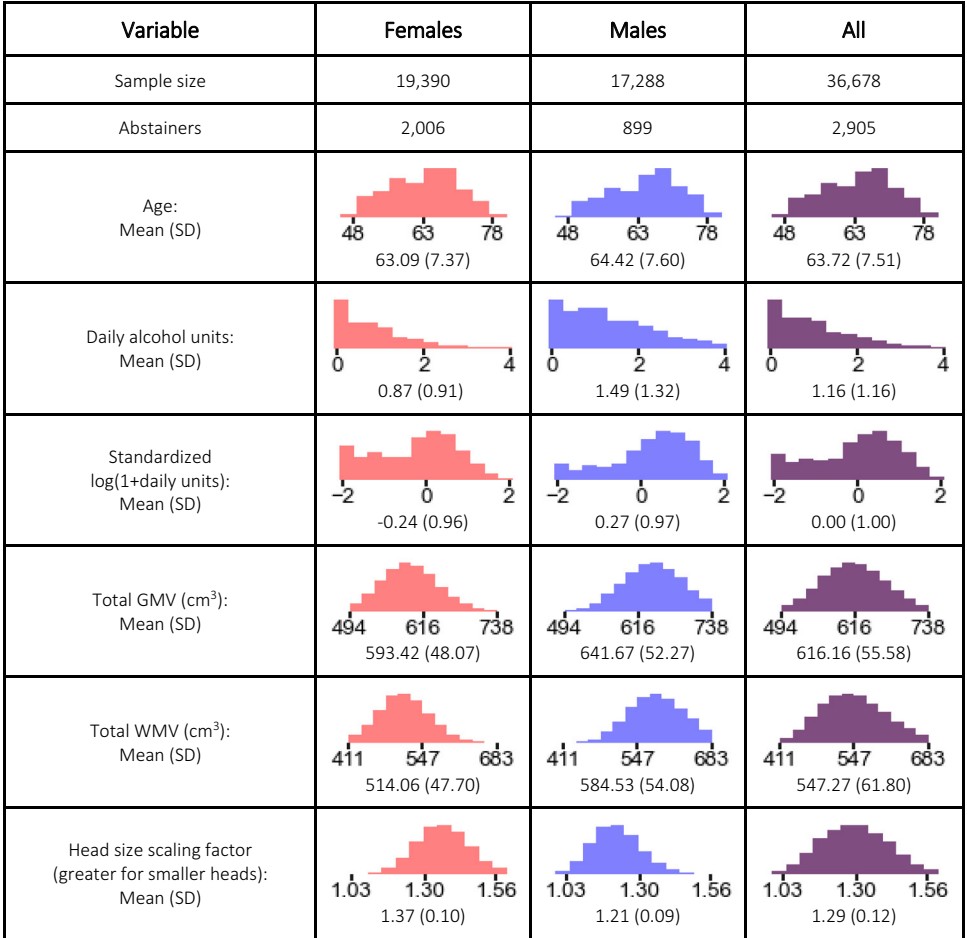

**Fig. 1 Empirical distributions of variables.** SD standard deviation, GMV gray matter volume, WMV white matter volume.

sex. The figures include local polynomial regression lines (LOWESS), which indicate negative trends on every dimension. This preliminary analysis also demonstrates slight non-linearities in both dimensions, with curves appearing concave. Therefore, our subsequent regression models include both linear and quadratic terms for age and logged alcohol intake, test the joint significance of the associations of the two terms with brain structure via an F-test (see "Methods"), and quantify the effect size via the IDP variance explained by alcohol intake, above the other covariates ($\triangle R^2$).

We estimate linear regressions to quantify the relationships between daily alcohol intake and its interactions with age, sex, and the global IDPs. Our primary analyses ($N = 36,585$) control for age, height, handedness, sex, smoking status, socioeconomic status, genetic ancestry, and county of residence (see Methods). Table 1 summarizes the results, revealing that both global IDPs decrease as a function of daily alcohol intake. Alcohol intake explains 1% of the variance in global GMV and 0.3% of the variance in global WMV across individuals beyond all other control variables (both $p < 10^{-16}$). Additional analyses excluding abstainers ($N = 33,733$) or excluding individuals who consume a high level of alcohol (i.e., females who report consuming more than 18 units/week and males who report consuming more than 24 units/week) ($N = 34,383$) and models using an extended set of covariates (including BMI, educational attainment, and weight; $N = 36,678$) yield similar findings, though the variance explained by alcohol intake beyond other control variables is reduced to 0.4% for GMV and 0.1% for WMV when individuals who

consume a high level of alcohol are excluded (Supplementary Tables 1 and 2).

In the eight regressions we tested, the interaction between alcohol intake and sex is not significant at the 1% level, except weakly for GMV when including the extended control variables (BMI, weight, and educational attainment). Similarly, none of the interactions between alcohol intake and age are significant at the 10% level. Only the regressions that exclude abstainers were significant at the 10% level ($p = 0.034$ for GMV, $p = 0.0014$ for WMV). Consequently, we excluded the interaction terms from the analyses of local IDPs.

We further used our regression models to calculate the predicted change in global GMV and WMV associated with increasing daily alcohol intake by one unit (Table 2). This prediction is similar when using different sets of control variables and when excluding individuals who did not consume alcohol or those who consume a high level of alcohol (for illustration, the change resulting from increasing alcohol intake from zero to one daily unit results in a reduction of $-0.030$ SD in global GMV and $-0.020$ SD in global WMV in the model estimated on the full sample and the model that excludes individuals who consume a high level of alcohol yields negative associations of similar magnitudes: $-0.034$ SD in global GMV and $-0.019$ SD in WMV. Given the non-linear relationship between global IDPs and alcohol intake, the associations vary across the drinking range. The change in the predicted global GMV and WMV when shifting from no daily alcohol consumption to one daily alcohol unit was less than 0.03 standard deviations. However, the

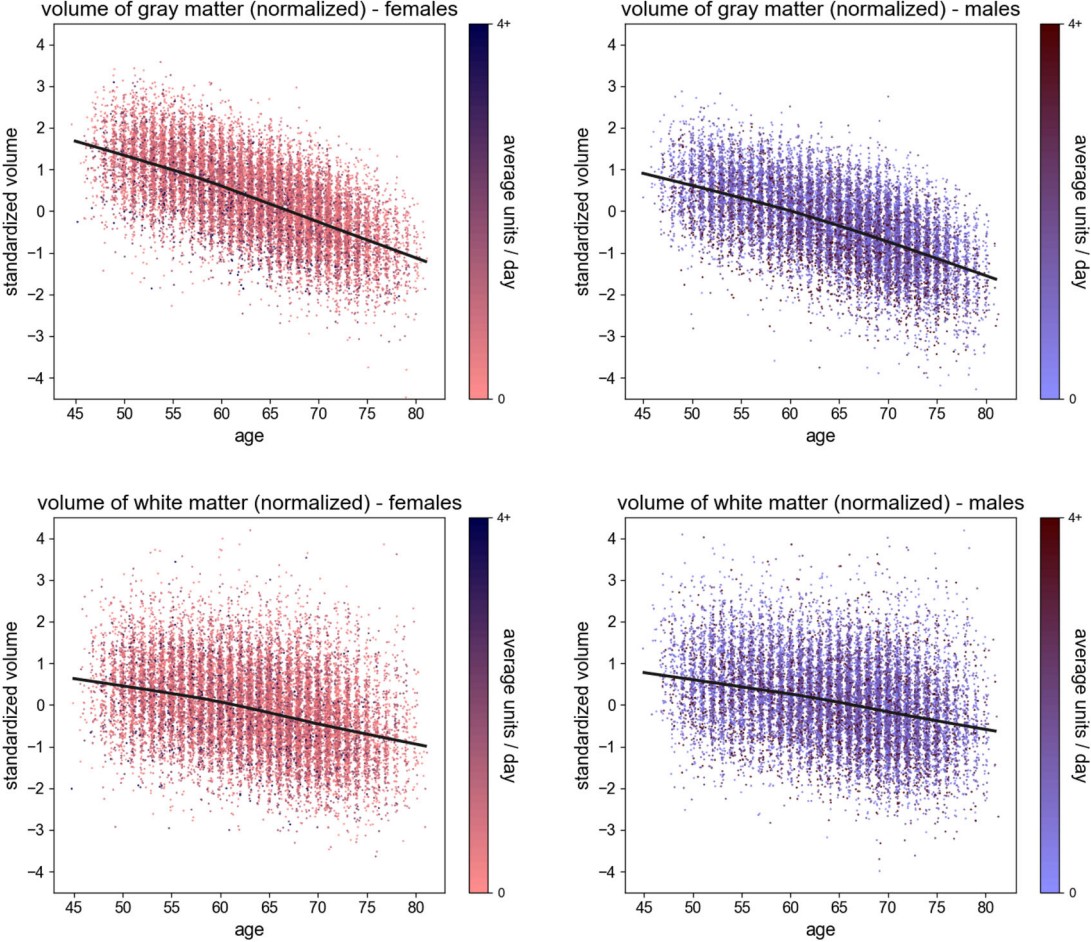

**Fig. 2 Scatter plots of whole-brain standardized gray matter volume (females, upper left; males, upper right) and standardized white matter volume (females, lower left; males, lower right), all normalized for head size, against the individual's age (*x*-axis).** The plots also show the LOWESS regression line (smoothness: $a = 0.2$). The 95% confidence interval is indistinguishable from the regression line. The colors are representative of the average daily alcohol consumption.

observed associations between alcohol intake and global IDPs increase as the number of daily units increases. An increase from one to two daily units is associated with a decrease of 0.127 and 0.074 standard deviations in predicted global GMV and WMV, respectively. A change from two to three daily units is associated with a 75% greater decrease of 0.223 and 1.28 standard deviations in GMV and WMV, respectively. Table 3 benchmarks the predicted effect magnitudes against the effects associated with aging for an average 50-year-old UKB participant, based on our regression models in the full sample. Table 4 replicates these results using the models that exclude individuals who consume a high level of alcohol. For illustration, the effect associated with a change from one to two daily alcohol units is equivalent to the effect of aging 2 years (or 1.7 years in the model that excludes individuals who consume a high level of alcohol), where the increase from two to three daily units is equivalent to aging 3.5 years (or 2.9 years in the model that excludes individuals who consume a high level of alcohol).

Figure 4 displays the averages of the two global IDPs in sub-samples binned according to their daily alcohol consumption range, illustrating the non-linear nature of the relationship between daily units and the global measures. The figure includes statistical tests that compare the average of the IDPs in the different sub-samples to their average in participants who consume one daily unit or less. These tests identify statistically significant associations for all bins of participants consuming

more than one daily unit, including those consuming 1–2 daily units. These associations are observed both in the full sample and within sex. Supplementary Figure 1 replicates Fig. 4 with the exclusion of individuals who consume a high level of alcohol, similarly demonstrating significant associations in participants who consume 1–2 units daily.

**Relationship between regional GMV and alcohol intake**. To investigate whether the reduction in global GMV associated with alcohol intake stems from relationships in specific regions, we estimate regression models to quantify the association of alcohol intake with a total of 139 regional GMV IDPs. These IDPs were derived using parcellations from the Harvard–Oxford cortical and subcortical atlases and Diedrichsen cerebellar atlas. Of the 139 GMV IDPs, 125 (88.9%) are significantly associated with log alcohol intake (see Supplementary Table 3). We observe the strongest associations in frontal, parietal, and insular cortices, temporal and cingulate regions, putamen, amygdala, and the brain stem. In these regions, alcohol intake explains between 0.3 and 0.4% of the variance in local GMV above the other covariates. Supplementary Figure 2 illustrates the marginal effect of increasing daily alcohol units on regional GMV IDPs, grouped by lobe. All associations are negative, except that involving the right pallidum—where the effect size is positive but very small ($\triangle R^2 = 0.0005$). Importantly, the largest regional association was

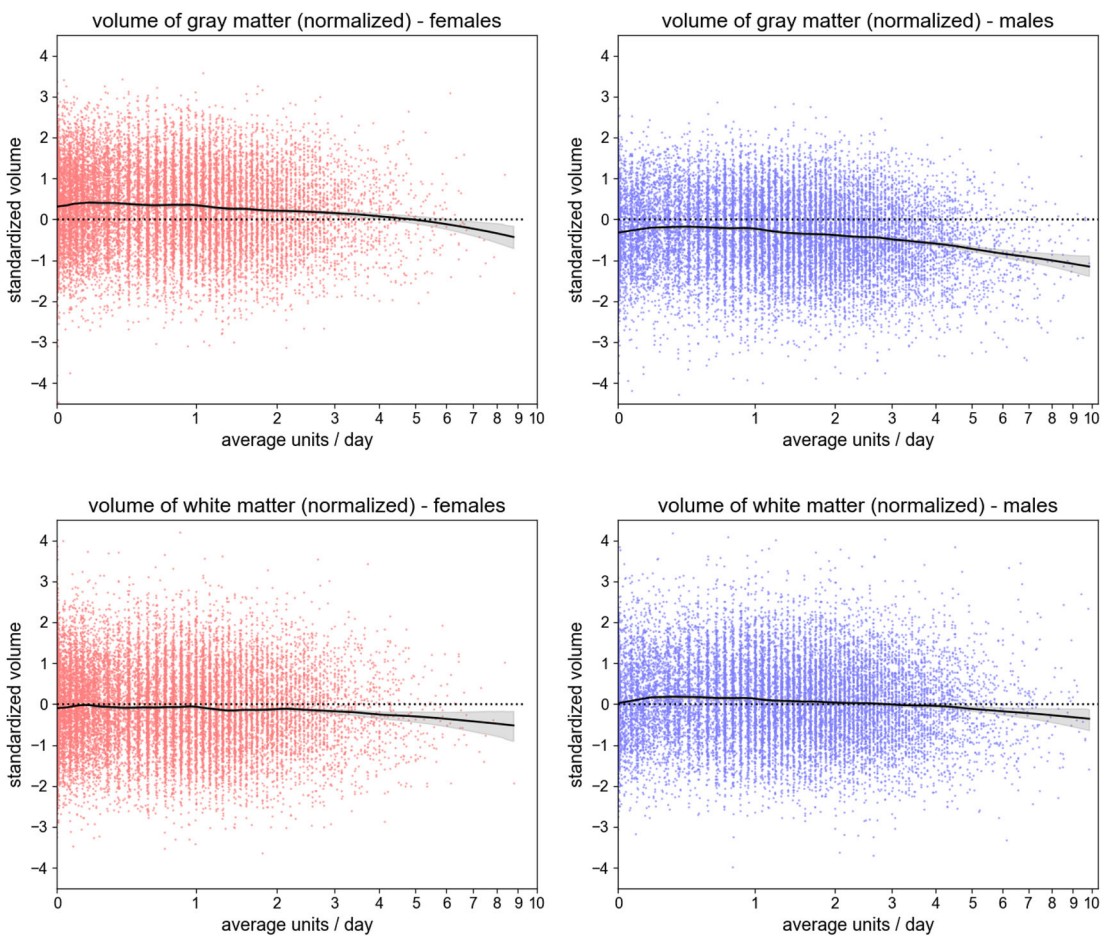

**Fig. 3 Scatter plots of whole-brain standardized gray matter volume (females, upper left; males, upper right) and standardized white matter volume (females, lower left; males, lower right), all normalized for head size, against the individual's daily alcohol consumption (x-axis, in log scale).** The plots also show the LOWESS regression line (smoothness: $a = 0.2$), with its 95% confidence interval. The dashed line represents the average standardized volume of the full sample (males and females).

**Table 1 Regression analysis with global IDPs as outcome variables.**

| Variable | Dependent variable: global GMV | | Dependent variable: global WMV | |
|---|---|---|---|---|
| | **N: 36,678 (df: 36,585), $R^2$: 0.514** | | **N: 36,678 (df: 36,585), $R^2$: 0.514** | |
| | **Regression coefficient (SE), 95% CI** | **t-stat (p-value)** | **Regression coefficient (SE), 95% CI** | **t-stat (p-value)** |
| Intake | −0.1095 (0.0058), CI: [−0.1209, −0.0982] | −19.0 ($p < 1.0e{-}16$) | −0.0650 (0.0078), CI: [−0.0802, −0.0498] | −8.4 ($p < 1.0e{-}16$) |
| Intake² | −0.0651 (0.0037), CI: [−0.0723, −0.0579] | −17.7 ($p < 1.0e{-}16$) | −0.0370 (0.0050), CI: [−0.0468, −0.0273] | −7.5 ($p = 7.8e{-}14$) |
| Intake × male | 0.0174 (0.0080), CI: [0.0018, 0.0330] | 2.2 ($p = 2.9e{-}02$) | 0.0164 (0.0107), CI: [−0.0046, 0.0374] | 1.5 ($p = 1.2e{-}01$) |
| Intake × std. age | 0.0080 (0.0037), CI: [0.0008, 0.0152] | 2.2 ($p = 3.0e{-}02$) | 0.0111 (0.0050), CI: [0.0014, 0.0208] | 2.2 ($p = 2.5e{-}02$) |
| Std. age | −0.5991 (0.0038), CI: [−0.6066, −0.5916] | −157.0 ($p < 1.0e{-}16$) | −0.3213 (0.0051), CI: [−0.3313, −0.3112] | −62.6 ($p < 1.0e{-}16$) |
| Std. age² | −0.0378 (0.0034), CI: [−0.0445, −0.0311] | −11.0 ($p < 1.0e{-}16$) | −0.0127 (0.0046), CI: [−0.0217, −0.0037] | −2.8 ($p = 5.7e{-}03$) |
| | Against model without intake and interactions Delta $R^2$: 0.0099, F-test: $p < 1.0e{-}16$ | | Against model without intake and interactions Delta $R^2$: 0.0033, F-test: $p < 1.0e{-}16$ | |

Note. *IDPs* imaging derived phenotypes, *GMV* gray matter volume, *WMV* white matter volume, *df* degrees of freedom, *SE* standard error, *CI* confidence interval, *std. age* standard age. All regressions include standard control variables. Intake is measured in log(1 + daily units of alcohol).

**Table 2 Predicted average additional effect (in standard deviations of IDP) of increasing alcohol intake by one daily unit on whole-brain gray matter volume and white matter volume, for models with different sets of control variables (first and second columns), and for standard control variables with samples excluding abstainers (third column) and individuals who consume a high level of alcohol (last column).**

| Intake changes | Standard control variables | | Extended control variables | | Excluding abstainers | | Excluding those that consume a high level of alcohol | |
|---|---|---|---|---|---|---|---|---|
| | Global GMV | Global WMV | Global GMV | Global WMV | Global GMV | Global WMV | Global GMV | Global WMV |
| 0–1 unit | −0.030 | −0.020 | −0.038 | −0.017 | −0.019 | −0.015 | −0.034 | −0.019 |
| 1–2 units | −0.127 | −0.074 | −0.126 | −0.073 | −0.123 | −0.070 | −0.107 | −0.067 |
| 2–3 units | −0.223 | −0.129 | −0.214 | −0.129 | −0.226 | −0.124 | −0.181 | −0.116 |
| 3–4 units | −0.319 | −0.184 | −0.302 | −0.185 | −0.330 | −0.179 | −0.255 | −0.164 |
| 0–4 units | −0.699 | −0.407 | −0.682 | −0.404 | −0.699 | −0.388 | −0.577 | −0.367 |

Note. *IDP* imaging derived phenotype, *GMV* gray matter volume; *WMV* white matter volume.

**Table 3 Predicted equivalent effect of aging in terms of additional years for an average 50-year-old individual (model includes all participants).**

| Intake changes | Standard control variables | | | |
|---|---|---|---|---|
| | Global GMV | Equivalent aging at 50 | Global WMV | Equivalent aging at 50 |
| 0–1 unit | −0.030 | 0.5 years | −0.020 | 0.5 years |
| 1–2 units | −0.127 | 2.0 years | −0.074 | 2.0 years |
| 2–3 units | −0.223 | 3.5 years | −0.129 | 3.5 years |
| 3–4 units | −0.319 | 4.9 years | −0.184 | 4.9 years |
| 0–4 units | −0.699 | 10.2 years | −0.407 | 10.4 years |

Note. *GMV* gray matter volume.

**Table 4 Predicted equivalent effect of aging in terms of additional years for an average 50-year-old individual (model excludes individuals who consume a high level of alcohol (i.e., females who report consuming more than 18 units/week and males who report consuming more than 24 units/week).**

| Intake changes | Standard control variables | | | |
|---|---|---|---|---|
| | Global GMV | Equivalent aging at 50 | Global WMV | Equivalent aging at 50 |
| 0–1 unit | −0.034 | 0.5 years | −0.019 | 0.5 years |
| 1–2 units | −0.107 | 1.7 years | −0.067 | 1.8 years |
| 2–3 units | −0.181 | 2.9 years | −0.116 | 3.1 years |
| 3–4 units | −0.255 | 4.0 years | −0.164 | 4.4 years |
| 0–4 units | −0.577 | 8.6 years | −0.367 | 9.5 years |

Note. *GMV* gray matter volume.

less than half the size of the association between alcohol consumption and global GMV, indicating that the global reduction in GMV associated with alcohol intake results from aggregating smaller associations that are widespread across the brain (rather than constrained to specific areas).

In a similar fashion to the analysis using the global IDPs, we calculate the average localized GMV IDP for each daily alcohol unit bin (Supplementary Fig. 3) and test their difference against the average of the group drinking up to one unit per day, within sexes and in the overall sample. As expected, the number of regional GMV IDPs showing a significant negative association with alcohol intake, and the magnitude of these associations increase as the average number of daily alcohol units increases. There are few regions (e.g., fusiform cortex) where lower GMV is either not observed as a function of alcohol intake or only apparent among individuals who consume a high level of alcohol (i.e., females who report consuming more than 18 units/week and males who report consuming more than 24 units/week). However, in most regions, GMV reduction is already visible in the groups that report moderate alcohol consumption (i.e., consuming 1–2 or 2–3 daily units). Thus, the association between moderate alcohol intake and GMV appears widespread across the brain and is detectable in males and females.

**Relationship between regional WM microstructure and alcohol intake**. To evaluate the relationship between alcohol intake and different indicators of WM integrity at the regional level, we estimate linear regressions to quantify the association of alcohol intake with 375 IDPs, including FA, MD, ICVF, ISOVF, and OD measures extracted via averaging parameters across 74 WM tract regions[45]. Of the 375 WM microstructure IDPs, 179 (47.7%) are

significantly associated with alcohol intake (see Supplementary Table 4). Generally, alcohol intake is related to lower coherence of water diffusion, lower neurite density, and higher magnitude of water diffusion, indicating less healthy WM microstructure with increasing alcohol intake.

To visualize the magnitude of WM microstructure IDP associations with alcohol intake, Fig. 5 displays the statistically significant and non-significant associations, alongside the average change in normalized WM microstructure IDPs associated with a mean daily alcohol intake increasing from 1 to 2 units (for a similar visualization of the changes associated with the increase from 2 to 3 units, see Supplementary Fig. 4). Thirteen WM tract regions show consistent associations with lower FA and higher ISOVF and MD. The strongest of these are in the fornix, where WM integrity was previously found to be associated with alcohol intake in studies of populations with AUD[3,21,23]. In the fornix, alcohol intake accounts for 0.45% of the variance in ISOVF, 0.35% of the variance in MD, and 0.32% of the variance in FA. Other WM tract regions showing a similar pattern yet with associations of weaker magnitude include commissural fibers (genu and body of the corpus callosum, bilateral tapetum), projection fibers (bilateral anterior corona radiata), associative fibers (fornix cres+stria terminalis, left inferior longitudinal fasciculus), and the bilateral anterior thalamic radiations.

Among the NODDI measures, ISOVF showed the strongest associations of alcohol intake all over the brain, most notably in the tract regions discussed above. The associations between alcohol intake and ICVF are also consistently negative yet smaller

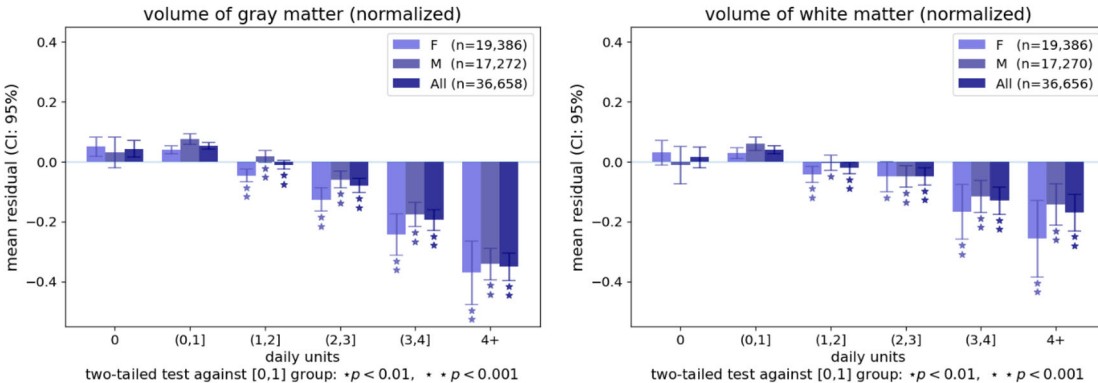

**Fig. 4 Bar plots representing the average residual volume of whole-brain gray and white matter volume for individuals grouped by the number of daily alcohol units after controlling for standard control variables.** The mean residuals are in terms of standard deviations of the dependent variable, where zero represents the average residual in the full sample. The error bars represent the 95% confidence interval. *$p < 0.01$ and **$p < .0001$ for groups showing a significant difference against the group consuming up to one alcohol unit daily. Supplementary Figure 1 replicates this Figure following the exclusion of individuals who consume a high level of alcohol (i.e., females who report consuming more than 18 units/week and males who report consuming more than 24 units/week).

in size. Daily alcohol intake explains no more than 0.1% of the variance beyond other control variables in all ICVF IDPs.

## Discussion

We report a multimodal brain imaging study of 36,678 middle-aged and older adults of European descent, a population sample whose reported alcohol consumption ranged from low (i.e., 1–2 alcohol units per day) to high (i.e., more than 4 alcohol units per day) levels of intake. The scale and granularity of the data provide ample statistical power to identify small associations while accounting for important potential confounds. We observe negative relationships between alcohol intake and global gray and white matter measures, regional GMVs, and WM microstructure indices. The associations we identify are widespread across the brain, and their magnitude increases with the average absolute number of daily alcohol units consumed.

Notably, the negative associations we observe with global IDPs are detectable in individuals who consume between 1 and 2 alcohol units daily. Thus, in the UK, consuming just one alcoholic drink daily (or two units of alcohol) could be associated with changes in GMV and WMV in the brain.

The negative associations between alcohol intake and total GMV and WMV are consistent with prior studies of early middle-aged[46] and older adults[28,47]. Based on Figs. 2 and 3, which show that males consumed more alcohol units per day and had larger global GMV and WMV, we further examine the influence of sex in detail. We find negative associations between alcohol intake and the global IDPs for both sexes and weak evidence for interactive effects between alcohol intake and sex on the brain. These findings are similar to a recent study of early middle-aged adult moderate drinkers that showed smaller brain volumes associated with moderate alcohol consumption in men and women[46]. The weak sex-by-alcohol interactions also comport with the findings of an earlier longitudinal study in individuals with AUD[38]; however, other cross-sectional studies have reported greater volume changes in women than men[48,49].

Although nearly 90% of all regional GMVs show significant negative associations with alcohol intake, the most extensively affected regions included the frontal, parietal, and insular cortices, with changes also in temporal and cingulate regions. Associations are also marked in the brain stem, putamen, and amygdala. The share of variance explained by alcohol intake for these regions is smaller in size than for global GMV, suggesting that the latter

results from an aggregation of many small effects that are widespread, rather than a localized effect that is limited to specific regions. Alcohol intake is further associated with poorer WM microstructure (lower FA and higher ISOVF and MD) in specific classes of WM tract regions. The commissural fibers (genu and body of the corpus callosum, bilateral tapetum), projection fibers (bilateral anterior corona radiata), associative bundles (fornix, fornix cres+stria terminalis, left inferior longitudinal fasciculus), and the bilateral anterior thalamic radiations show the most consistent associations with alcohol intake, with the fornix showing the strongest associations. The fornix is the primary outgoing pathway from the hippocampus[50], and WM microstructural alterations in the fornix are consistently associated with heavy alcohol consumption and memory impairments[3,51].

Our findings are partly consistent with studies of individuals with AUD[18,52]. The pattern of microstructural alterations in our general population sample show that widespread WM alterations are present across multiple WM systems. Like individuals with AUD, alcohol intake in this healthy population sample is associated with microstructural differences in superficial WM systems functionally related to GM networks, including the frontoparietal control and attention networks, and the default mode, sensorimotor, and cerebellar networks. Deeper WM systems (superior longitudinal fasciculus and dorsal frontoparietal systems, inferior longitudinal fasciculus system, and deep frontal WM) thought to be involved in cognitive functioning by regulating reciprocal connectivity[52,53] are also associated with alcohol intake. Within these WM systems, alcohol intake is most strongly associated with ISOVF, MD, and FA WM microstructure indices; whereas, associations with ICVF are small, and OD associations are inconsistent or nonexistent. Alcohol intake shows positive associations with ISOVF and MD and negative associations with FA. This pattern of alcohol-associated WM microstructural disruption supports previous research showing excessive intracellular and extracellular fluid in individuals with AUD[20].

Our study has several limitations. First, we rely on a sample of middle-aged individuals of European ancestry living in the UK. We hope that future work will test the generalizability of our findings to individuals from other populations and in other age groups. It is reasonable to expect that the relationship we observe would differ in younger individuals who have not experienced the chronic effects of alcohol on the brain. An additional limitation stems from the self-reported alcohol intake measures in the UK Biobank, which cover only the year prior to participation. Such

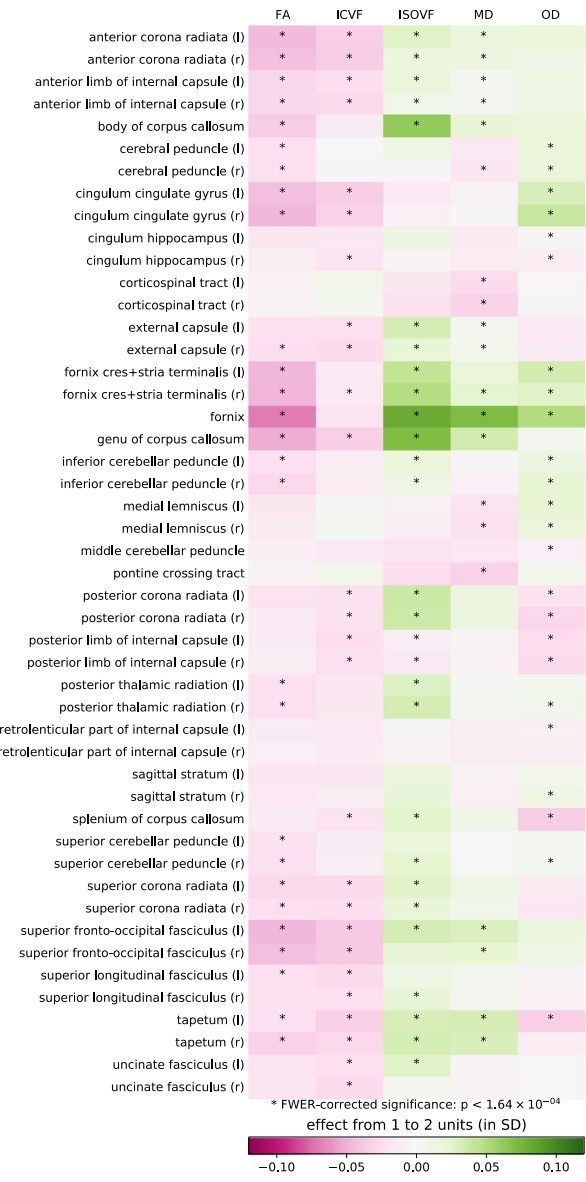

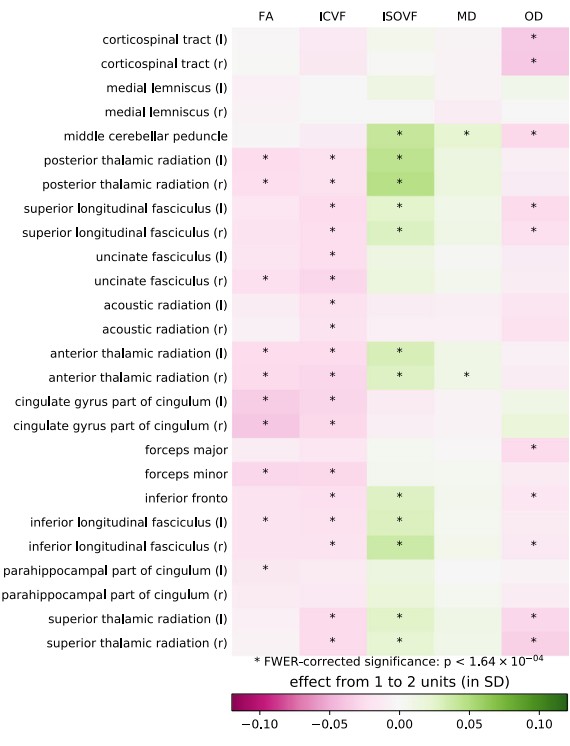

**Fig. 5 Associations between daily alcohol units and white matter microstructure indices of interest across white matter tract regions.** Asterisks denote statistically significant effects, $p < 1.64 \times 10^{-4}$. Colors represent the expected change in each imaging derived phenotype resulting from the increase in daily consumption from 1 to 2 units, based on the regression model. FA fractional anisotropy, ICVF intracellular volume fraction, ISOVF isotropic volume fraction, MD mean diffusivity, OD orientation dispersion, r right, l left.

estimates may not adequately reflect drinking prior to the past year and are susceptible to reporting and recall bias[38,39].

Further, our analyses do not account for individuals with a past diagnosis of AUD. Earlier studies have shown that the brain shows some recovery with the cessation of drinking in individuals with AUD, but this varies with age and sex, and recovery might be incomplete[54–56]. Thus, a past diagnosis of AUD would likely influence our results. We hope future studies will shed light on how a history of AUD with prolonged recovery is associated with brain structure in middle-aged and older adults. Moreover, partial volume effects (e.g., voxels containing cerebrospinal fluid (CSF)) can reduce the accuracy of tissue characterization and WM microstructural estimates. Previous research indicates that partial volume effects can bias diffusion measures toward a pattern of high diffusivity (MD) and reduced FA, particularly in intraventricular locations like the fornix[57,58]. As such, our findings could reflect partial volume effects; however, it should be noted that the structural data were acquired using T2-weighted FLAIR imaging,

a structural technique that mitigates CSF contamination by suppressing signal from fluid (i.e., CSF).

Finally, our study relies on a cross-sectional design, which does not allow for the identification of causal effects. While our models account for more potential confounding variables than earlier observational studies in this area of research, we cannot rule out the possibility of reverse-causality or a confounding influence of other factors that are not included in our models. Further investigation of the causal nature of the relationships between alcohol intake and brain anatomy (e.g., via longitudinal studies or natural experiments) would be of interest.

In summary, this study provides additional evidence for a negative association between alcohol intake and brain macro-structure and microstructure in a general population sample of middle-aged and older adults. Alcohol intake is negatively associated with global brain volume measures, regional GMVs, and WM microstructure. The associations between alcohol intake and regional GMV are evident across the entire brain, with the largest

volume changes observed in frontal, parietal, and insular cortices, temporal and cingulate regions, the brain stem, putamen, and amygdala. Alcohol intake is related to WM microstructural alterations in several WM tract regions connecting large-scale networks and deeper WM systems. Most of these negative associations are apparent in individuals consuming an average of only one to two daily alcohol units. Thus, this multimodal imaging study highlights the potential for even moderate drinking to be associated with changes in brain volume in middle-aged and older adults.

## Methods

**Sample, procedure, and exclusion criteria**. Our sample comprised 36,678 individuals of European ancestry from the UKB, all study participants whose data were available as of September 1, 2020. All UK Biobank (www.ukbiobank.ac.uk) participants provided written informed consent, and the North West Multi-Center Ethics committee granted ethical approval. Participants provided demographic and health information via touchscreen questionnaires. A nurse conducted a medical history interview, which included self-report of medical diagnoses and other conditions or life events that were used to evaluate eligibility to participate (study details are available at https://www.ukbiobank.ac.uk/media/gnkeyh2q/study-rationale.pdf). Vital signs were obtained, and BMI was calculated as weight (kg)/height$^2$ (m).

The data was provided by the UK Biobank and was already subject to quality control[59]. We excluded individuals with IDP values outside a range of four standard deviations (SDs). Given the large sample size, we chose this lenient threshold as a non-trivial number of observations (97 for GM, 127 for WM) fall between three and four SDs away from the mean. The IDPs beyond the four SD range are likely the results of processing errors, or the corresponding individuals present severe brain irregularities (5 individuals for GM, 7 for WM). Note that excluding these outliers does not change the statistical significance or magnitude of our reported effects. The exclusion of individuals falling within three SDs of the mean does not change the results either.

**Measures of alcohol consumption**. Participants self-reported the number of alcohol units (10 ml of pure ethanol) consumed, in "units per week" (for frequent drinkers) or "units per month" (for less frequent drinkers), across several beverage categories (red wine, white wine/champagne, beer/cider, spirits, fortified wine, and "other"). The UKB assessment defined units of alcohol as follows: a pint or can of beer/lager/cider = two units; a 25 ml single shot of spirits = one unit; and a standard glass of wine (175 ml) = two units. We computed the number of weekly units by summing the weekly units consumed in all categories. When reported monthly, the intake was converted to units per week by dividing by 4.3. The number of weekly units was divided by 7 to determine units per day.

**MRI data acquisition and processing**. MRI data were acquired using a Siemens Skyra 3T scanner (Siemens Healthcare, Erlangen, Germany) using a standard 32-channel head coil, according to a freely available protocol (http://www.fmrib.ox.ac.uk/ukbiobank/protocol/V4_23092014.pdf). As part of the scanning protocol, high-resolution T1-weighted images, three-dimensional T2-weighted fluid-attenuated inversion recovery (FLAIR) images, and diffusion data were obtained. High-resolution T1-weighted images were obtained using an MPRAGE sequence with the following parameters: TR = 2000 ms; TE = 2.01 ms; 208 sagittal slices; flip angle, 8°; FOV = 256 mm; matrix = 256 × 256; slice thickness = 1.0 mm (voxel size 1 × 1 × 1 mm); total scan time = 4 min 54 s. 3D FLAIR images were obtained with the following parameters: TR = 1800 ms; TE = 395.0 ms; 192 sagittal slices; FOV = 256 mm; 256 × 256; slice thickness = 1.05 mm (voxel size 1.05 × 1×1 mm); total scan time = 5 min 52 s. Diffusion acquisition comprised a spin-echo echo-planar sequence with 10 T2-weighted ($b ≈ 0$ s mm$^{-2}$) baseline volumes, 50 b = 1000 s mm$^{-2}$ and 50 b = 2000 s mm$^{-2}$ diffusion-weighted volumes, with 100 distinct diffusion-encoding directions and 2 mm isotropic voxels; total scan time = 6 min 32 s.

Structural imaging and diffusion data were processed by the UK Biobank team and made available to approved researchers as imaging-derived phenotypes (IDPs); the full details of the image processing and QC pipeline are available in an open-access article[42,60]. IDPs used in analyses included whole-brain GMV, whole-brain WMV, 139 regional GMV IDPs derived using parcellations from the Harvard–Oxford cortical and subcortical atlases and Diedrichsen cerebellar atlas (UKB fields 25782–25920), and 375 tract-averaged measures of FA, MD, ICVF, ISOVF, and orientation diffusion (OD) extracted by averaging parameters over 74 different white-matter tract regions based on subject-specific tractography[61] and from population-average WM masks[45]. Volumetric IDPs were normalized for head size by multiplying the raw IDP by the T1-based "head size scaling factor"[60].

## Statistical analyses

*Descriptive analysis using global IDPs.* We plot global GMV and WMV in males and females separately, normalized for head size, against age (Fig. 2) and alcohol intake (i.e., alcohol units/day on a log scale) (Fig. 3).

*Global IDPs, regional GMV, and WM microstructure analyses.* Our primary analysis estimates a linear regression of several IDPs on alcohol intake in log(1 + daily units), including various control variables and interactions. Given the slight concavity of the LOWESS regression lines in the descriptive analysis of the global IDPs, we included both linear and quadratic terms for alcohol intake and age in the regression:

$$IDP_i = \beta_0 + \beta_1 X_i + \beta_2 X_i^2 + \beta_3 X_i \times SEX_i + \beta_4 X_i \times AGE_i + \gamma Z_i + e_i,$$

where $IDP_i$ is the IDP normalized for head size, $X_i$ is the standardized alcohol intake in log(1 + daily units), $AGE_i$ is standardized age, $Z_i$ is a vector of control variables, and $e_i$ is an error term.

Our analyses comprise models that include two different sets of control variables. The standard set includes standardized age, standardized age squared standardized height, handedness (right/left/ambidextrous; dummy-coded), sex (female:0, male:1), current smoker status, former light smoker, former heavy smoker, and standardized Townsend index of social deprivation measured at the zip code level[62]. To control for genetic population structure, the models also include the first 40 genetic principal components[63] and county of residence (dummy-coded)[62]. A second set of extended control variables includes all standard control variables and in addition standardized BMI, standardized educational attainment[64], and standardized weight. To determine whether observations at the extreme ends of the drinking distribution bias the estimates of the relationship between alcohol intake and IDPs, we also estimate a model that excludes abstainers and a model that excludes heavy drinkers (i.e., women who reported consuming more than 18 units/week and men who consumed more than 24 units/week), both with standard controls. For each IDP, we test the null hypothesis that alcohol did not affect the outcome measure via an F-test that compares our model against a model with only the control variables (excluding alcohol intake and related interaction terms).

We separate the analysis into two parts: (1) global analysis and (2) regional GMV and WM microstructure analysis, including 514 IDPs in total (139 GMV IDPs, 375 WM microstructure IDPs). The interactions of alcohol intake variables with sex and age are not significant in the global analysis ($p > 0.001$), so we exclude them from the regional analyses. To control the family-wise error rate in the regional GMV and WM microstructure analysis, we determine the significance thresholds for all regressions using the Holm method[65], ensuring a family-wise error rate below 5%. When testing for $M$ hypotheses, this method orders the corresponding p-values from lowest to highest: $p_0,..., p_M$, and identifies the minimal index $k$ such that $p_k > 0.05/(M + 1 - k)$. All hypotheses with an index $m < k$ are then considered to be statistically significant. In our application, the significance threshold was determined to be $1.64 × 10^{-4}$.

To quantify and visualize associations between alcohol intake and IDPs (i.e., global GMV and WMV, and regional GMV IDPs), we bin participants in the following six categories based on average alcohol intake: (1) abstainers, (2) individuals who drank less than one unit/day, (3) individuals who drank between one (included) and two (excluded) units/day (recommended maximal alcohol consumption based on the UK Chief Medical Officers "low-risk" guidelines[32]), (4) individuals who drank between two (included) and three (excluded) units/day, (5) individuals who drank between three (included) and four (excluded) units/day, and (6) individuals who drank at least four units/day. After regressing the influence of the standard control variables, we then calculate the mean residual values (measured in standard deviations of IDPs) and 95% confidence intervals (CI). By first regressing the dependent variables on the standard control variables, the estimated effect can be interpreted as the part of the change in IDP that is not explained by these other variables, and it is represented in terms of standard deviations from the average. Results are available to the readers in supplementary data figures and tables. For example, Supplementary Fig. 3 includes the average GMV of all regions tested (both significant and non-significant) in bins of participants with different daily alcohol intake levels.

*Pre-registration.* We registered the analysis plan was preregistered with the Open Science Foundation (https://osf.io/trauf/).

**Reporting summary**. Further information on research design is available in the Nature Research Reporting Summary linked to this article.

## Data availability
Data and materials are available via UK Biobank at http://www.ukbiobank.ac.uk/.

## Code availability
The analysis code used in this study is publicly available with the Open Science Framework (https://osf.io/trauf/?view_only=a3795f76c5a54830b2ca443e3e07c0f0).

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

## Acknowledgements

This research was carried out under the auspices of the Brain Imaging and Genetics in Behavioral Research Consortium (https://big-bear-research.org/), using UK Biobank resources under application 40830. The study was supported by funding from an ERC Consolidator Grant to PK (647648 EdGe), NSF Early Career Development Program grant (1942917) to GN, the National Institute on Alcohol Abuse and Alcoholism to RRW (K23 AA023894), and the VISN 4 Mental Illness Research, Education and Clinical Center at the Crescenz VA Medical Center. GN thanks Carlos and Rosa de la Cruz for their ongoing support.

## Author contributions

R.D., P.K., H.R.K., G.N., and R.R.W. conceived and designed the study. R.D. analyzed data. R.D., G.A., K.J., P.K., H.R.K., G.N., and R.R.W. interpreted data. R.D., G.N., and R.R.W. wrote the paper. G.A., N.S., P.K., and H.R.K. critically edited the work. R.D., G.N., and R.R.W. finalized all edits. All authors approved the final version to be submitted for publication and agree to be accountable for all aspects of this work.

## Competing interests

H.R.K. is a member of an advisory board for Dicerna Pharmaceuticals, Sophrosyne Pharmaceuticals, and Entheon Pharmaceuticals; a consultant to Sobrera Pharmaceuticals; a member of the American Society of Clinical Psychopharmacology's Alcohol Clinical Trials Initiative, which was supported in the last three years by AbbVie, Alkermes, Dicerna, Ethypharm, Indivior, Lilly, Lundbeck, Otsuka, Pfizer, Arbor, and Amygdala Neurosciences; and is named as an inventor on PCT patent application #15/878,640 entitled: "Genotype-guided dosing of opioid agonists," filed January 24, 2018. All other authors declare no competing interests.
