## [Peer Review File · Nature Communications]

Associations between alcohol consumption and gray and white matter volumes in the UK BiobankREVIEWER COMMENTS

Reviewer #1 (Remarks to the Author):

The sex distributions in the full population age fairly equal (48%male/52%female) but when stratified by alcohol consumption they are substantially unequal (abstainers 69% female, heavy drinkers 60.2% male). Thus, there is an interaction between sex and alcohol consumption. While the analyses controlled for age, sex and head size, sex and head size are confounded. The authors note in their introduction the role of sex and age in brain findings in alcohol abuse disorders, but the presentation here does not discuss any age or sex interactions with alcohol consumption on the brain measures. The literature review on alcohol effects on the brain is sparse and could be more informative. It appears that heavy drinking was defined as 4+ drinks/day. This cut-point, of course, would include a large percentage of people with alcohol use disorder. The authors note that their results are “predominantly driven by heavy drinkers” but “effects were also observed among individuals who reported consuming two units/day of alcohol.” Was the two units/day effect significant on its own or only when included in the regression analysis?

What are the error bars in the figures – standard errors or standard deviations?

The data are presented as regression residuals so the reader has no notion of the magnitude of the effects. The authors note that the effects they found were “small but significant.” This is the strength and weakness of large sample sizes – highly significant but small effects are revealed. How meaningful were the “significant” effects? How do those effects comport with the extensive (but not thoroughly referenced) literature on the effects of alcohol use disorder on the brain? How do they comport with other epidemiological studies that considered a wide range of drinking?

The authors contrast their approach, which sought linear to non-linear relations between alcohol consumption variables and brain macrostructural and microstructural integrity, with other published reports based on linear testing only. Yet, the findings in the figures do not show any indication of relations other than linear. Can the authors provide an explanation for the difference, or be clearer on noting where complex functions best describe relations?

Please provide scatterplots with regression functions by sex of the significant brain regions over the drinking variable and over age--distinguishing men and women by color, e.g., red for women and blue for men.

To be meaningful to the reader scatterplots of the brain values by drinking variable (which should not be quantized but be continuous) should show confidence interval (should be small given the large number of subjects) plus 1 and 2 standard deviations to demonstrate the variance.

The authors note a study limitation that the self-reported alcohol consumption covers only the past year, which would not account for people with past diagnosis of AUD. This shortcoming needs to be addressed, given that while the brain shows some recovery with prolonged sobriety, it can vary with age and sex and in fact may not recover completely.

The following two sentences appear to be in conflict--the last sentence of the Results and third sentence of the Discussion (page 14). Please clarify.

Can the authors provide a statement that relates to specific brain networks that appear to be most affected by drinking heavily? Are they the same in men and women and over age?

There is no discussion about the brain regions affected with regard to their representing brain systems likely affected by heavy drinking. In other words, the authors seek little in the way of neurobiological explanations of their findings. Further, they make no mention of having identified brain pathology despite the likelihood of having a large representation of Alcohol Use Disorder in the cohort.

Reviewer #2 (Remarks to the Author):

The current manuscript describes a novel analysis of data from the UK Biobank dataset. The authors examine grey matter and white matter volume, as well as more recent NODDI-based metrics of white matter streamline organization, in ~19,000 MRI samples. The strength of the manuscript comes from 2 main findings. First, all measures, including grey and white matter volume, FA, MD, and three

measures in most tracts show a decline with increasing alcohol consumption. Second, these declines were present independent of a large number of control variables. While the findings would have a clear significance to the study of alcohol-related brain changes, a general lack of theoretical and empirical dimensionality in these results suggests more work is necessary for future revisions. I outline this issue and a number of smaller issues below.

The general problem with the manuscript is that there is a lack of alternative hypotheses, analytical interpretation, or dimensionality in the current results. The reader is given essentially a very descriptive set of results, with no counterpoints. Non-significant results are important, but unreported or not visualized in the paper. All 7 measures are tested, but if there are different results across measures, these differences aren't tested statistically, and the authors make no conclusions about these different patterns across different white matter metrics. My suggestions in this respect would be to "dive deeper" in at least 3 different ways: (1) test for explicit regional differences (e.g., sensory vs. association regions, early vs. late myelinating regions), (2) explicit tests for the utility of different myelin metrics in detecting alcohol-related changes in the brain (are effects greater for ICVF than MD, reflecting greater changes in fiber organization vs overall diffusivity?), or (3) more explicit interaction tests for some of the many control variables interact (e.g., how does age interact with alcohol use in determining white matter health?, or are declines in WM with increasing alcohol consumption steeper in different socioeconomic groups?). The Biobank is a rich dataset, many interesting questions are possible!

A related, but more minor concern, is that the authors present a wealth of graphs/data, but it is poorly contextualized. I found myself searching through many different Supplementary documents to look for trends in the results, because these were not spelled out explicitly in the text. The figures in the main document aren't especially compelling, or visually easy to localize to specific brain regions. Perhaps the authors could reuse the images in Fig1 to help with this.

The authors regularly refer to "tracts", but these are more properly regions-of-interest (ROIs) within white matter regions—no tractography was performed in this study, and the inferences of canonical fiber systems are based on volumetric ROIs. As such, referring to "tracts" is misleading.

How many individuals are in each model? The authors start with 19,825, lose 747 due to data quality, and then in each of the several models run, the number of samples "decreased when phenotype data were missing", so it's unclear how many subjects contribute to each model. How many individuals were excluded based on IDP values greater than 4 standard deviations? This seems a rather lenient threshold for inclusion.

What motivation do the authors have for including 3 age-related variables (linear, 2nd, and 3rd order age effects)?

There are few details about how the authors apply the Holm method to their statistical testing.

Signed,
Simon Davis
Duke University

Reviewer #3 (Remarks to the Author):

This is an important study well-powered to examine effects of low to moderate alcohol consumption on brain structure, which has been a long-term gap in the literature. I find it compelling and it will be of broad interest to neuroimaging researchers given the recommendation that even low-moderate alcohol use impacts brain structure and should be included as a confound in structural analyses. My comments were primarily concerned with aspects of methodology and clarity of reporting the analyses and findings.

Introduction

Pg. 3: Your 'key questions' statements could be clearer, with careful reference to AUD versus the general population.

Pg. 4: When describing potential confounds, it would be useful to include a brief summary for each point e.g. sex (women more vulnerable than men). On this page you also state the number of WM tracts but not number of GM regions examined.

Results

Pg. 11: Per supplementary tables 2-3 please clarify that Model D resulted in no significant results other than for OD - this is an interesting finding to include in the main text given OD may then be the most sensitive WM measure of more subtle changes with lower exposure to neurotoxicity.

Numerous regions were examined, and while significant regions from Model A are reported in the supplementary figures and tables, an overall statement of significant regional results should be included somewhere for context (i.e., 16/139? (%) GM regions (pg 7?) and #/27 (%) WM tracts (pg 11?) were significantly associated with alcohol intake).

In the introduction you mention the key question of nonlinearity, but your analysis/reporting on that issue isn't very clear to me outside of using Models C and D and then forming the groups. Why wasn't there a direct examination of alcohol as nonlinear since the analysis is so well powered?

Discussion

Pg. 12: Please expand your discussion of regions of GM and WM with greatest effect sizes, and whether these regions are consistent with regions demonstrated in heavier drinkers (i.e., is it that there is strong overlap and by examining low-moderate drinkers we see regions that might be most vulnerable, or is there some other pattern with low-moderate than heavy drinkers?).

Methods

Pg. 15: Exclusions of IDPs more than 4 SDs from the mean seems very lenient considering all of the factors that influence brain structure, especially with aging. Why was 4 SD chosen rather than 3 SD? Do sample sizes or results significantly change with the additional exclusions?

Pg. 16: How skewed or otherwise non-normal was alcohol intake, assuming that underlies your log transformation?

Response to Reviewers

Thank you for the opportunity to revise and resubmit our manuscript NCOMMS-20-11997, now entitled “Multimodal brain imaging study of 36,678 participants reveals adverse effects of moderate drinking”. We genuinely appreciate the time and effort the reviewers dedicated to this manuscript, and thank you for the detailed feedback. The reviewers had excellent suggestions, which we have used to improve the manuscript substantially. The most notable changes that we made in this revision are the following:

(1) We increased the sample size to $N = 36,678$, thanks to a new release of brain images by the UK Biobank. The new sample is about double the previous sample size—which was already an order of magnitude greater than the largest single study on the topic to date. The increase in sample size boosted the statistical power of the study, which allowed us to test differences between subsamples of the population (e.g., by comparing individuals who consume 1-2 units daily to those who drink less than one unit, separately by sex). Because the substantial increase in statistical power enabled us to detect very small effects, we shifted the focus of our findings from statistical significance to the size of the effects, quantified as the variance in the IDPs explained by alcohol intake, above the control variables.

(2) We added scatter plots with local polynomial regression lines (LOWESS). These regressions characterize the relationships between age or alcohol intake and global measures of white matter and gray matter volumes, separately for males and females. These plots show slight concavity, and we therefore included both linear and quadratic values for alcohol intake and age in all of our regression models.

(3) We sought to increase the interpretability of our findings by putting them in context. For example, we make quantitative predictions for the changes in total gray and white matter volumes associated with differences in daily alcohol intake (e.g., between 0-1 or 1-2 units/day) and benchmarked them against age-related effects (Table 3).

(4) We revised the Introduction, Discussion, Tables and Figures based on the changes to the statistical approach and reviewers' comments.

We detailed these changes with point-by-point replies to each reviewer's specific comments (with the original comments in **boldface**) below.

Reviewer 1:

1. The sex distributions in the full population age fairly equal (48%male/52%female) but when stratified by alcohol consumption they are substantially unequal (abstainers 69% female, heavy drinkers 60.2% male). Thus, there is an interaction between sex and alcohol consumption. While the analyses controlled for age, sex and head size, sex and head size are confounded. The authors note in their introduction the role of sex and age in brain findings in alcohol abuse disorders, but the presentation here does not discuss any age or sex interactions with alcohol consumption on the brain measures.

Thank you for raising this important point. We agree that sex and age interactions with alcohol consumption are important to explore, and we address this issue in several ways.

1. For better contextualization, we provide empirical distributions of variables of interest for males, females, and the sample overall (Table 1 below).

Table 1. Empirical distributions of variables.

Variable	Women	Men	All
Sample size	19,390	17,288	36,678
Abstainers	2,006	899	2,905
Age: Mean (SD)	 48 63 78 63.09 (7.37)	 48 63 78 64.42 (7.60)	 48 63 78 63.72 (7.51)
Daily alcohol units: Mean (SD)	 0 2 4 0.87 (0.91)	 0 2 4 1.49 (1.32)	 0 2 4 1.16 (1.16)
Standardized log(1+daily units): Mean (SD)	 -2 0 2 -0.24 (0.96)	 -2 0 2 0.27 (0.97)	 -2 0 2 0.00 (1.00)
Total GMV (cm ³): Mean (SD)	 494 616 738 593.42 (48.07)	 494 616 738 641.67 (52.27)	 494 616 738 616.16 (55.58)
Total WMV (cm ³): Mean (SD)	 411 547 683 514.06 (47.70)	 411 547 683 584.53 (54.08)	 411 547 683 547.27 (61.80)
Head size scaling factor (greater for smaller heads): Mean (SD)	 1.03 1.30 1.56 1.37 (0.10)	 1.03 1.30 1.56 1.21 (0.09)	 1.03 1.30 1.56 1.29 (0.12)

Note. SD = standard deviation, GMV = gray matter volume, WMV = white matter volume.

2. We conducted descriptive analyses using scatter plots representing whole-brain gray matter volume and white matter volume against age and against average daily alcohol units, separately in males and females (Figures 2 and 3 below).

Figure 1. Scatter plots of whole-brain standardized gray matter volume (women, upper left; men, upper right) and standardized white matter volume (women, lower left; men, lower right), all normalized for head size, against the individual's age (x-axis). The plots also show the LOWESS regression line (smoothness: $a=0.2$). The 95% confidence interval is indistinguishable from the regression line. The colors are representative of the average daily alcohol consumption.

Figure 2. Scatter plots of whole-brain standardized gray matter volume (women, upper left; men, upper right) and standardized white matter volume (women, lower left; men, lower right), all normalized for head size, against the individual's daily alcohol consumption (x-axis, in log scale). The plots also show the LOWESS regression line (smoothness: $a=0.2$), with its 95% confidence interval.

3. We added age-alcohol and sex-alcohol interactions to the global gray matter volume and white matter volume regressions (Table 2 below). We found no significant effects of age-alcohol intake interactions on total gray matter volume. We found weak associations for age-alcohol interaction effects on total white matter volume only when abstainers were excluded. Similarly, we found weak sex-alcohol intake interaction effects across the eight whole-brain IDP regression models in the global gray matter volume regression only when both standard and extended (body mass index, educational attainment, weight) controls were included. Given the small interaction effects that could be detected and their p-values being consistently above 0.001, even in a sample of over 36,000 participants, we chose to exclude the interaction terms in subsequent analyses. We append the text referring to this finding (from the results section) below:

Table 2. Regression analysis with global IDPs as outcome variables. All regressions include standard controls. Intake is measured in $\log(1 + \text{daily units of alcohol})$.

Variable	Dependent variable: global GMV		Dependent variable: global WMV	
	N: 36,678 (d.f.: 36,585), R ² : 0.514		N: 36,678 (d.f.: 36,585), R ² : 0.514	
	Regression Coefficient (S.Err), 95% CI	t-stat (p-value)	Regression Coefficient (S.Err), 95% CI	t-stat (p-value)
intake	-0.1095 (0.0058), CI: [-0.1209,-0.0982]	-19.0 ($p < 1.0e-16$)	-0.0650 (0.0078), CI: [-0.0802,-0.0498]	-8.4 ($p < 1.0e-16$)
intake ²	-0.0651 (0.0037), CI: [-0.0723,-0.0579]	-17.7 ($p < 1.0e-16$)	-0.0370 (0.0050), CI: [-0.0468,-0.0273]	-7.5 ($p = 7.8e-14$)
intake x male	0.0174 (0.0080), CI: [0.0018,0.0330]	2.2 ($p = 2.9e-02$)	0.0164 (0.0107), CI: [-0.0046,0.0374]	1.5 ($p = 1.2e-01$)
intake x std. age	0.0080 (0.0037), CI: [0.0008,0.0152]	2.2 ($p = 3.0e-02$)	0.0111 (0.0050), CI: [0.0014,0.0208]	2.2 ($p = 2.5e-02$)
std. age	-0.5991 (0.0038), CI: [-0.6066,-0.5916]	-157.0 ($p < 1.0e-16$)	-0.3213 (0.0051), CI: [-0.3313,-0.3112]	-62.6 ($p < 1.0e-16$)
std. age ²	-0.0378 (0.0034), CI: [-0.0445,-0.0311]	-11.0 ($p < 1.0e-16$)	-0.0127 (0.0046), CI: [-0.0217,-0.0037]	-2.8 ($p = 5.7e-03$)
		Against model without intake and interactions Delta R ² : 0.0099, F-test: $p < 1.0e-16$	Against model without intake and interactions Delta R ² : 0.0033, F-test: $p < 1.0e-16$	

“In the eight regressions we tested, the interaction between alcohol intake and sex is not significant at the 1% level, except weakly for GMV when including the extended control variables (BMI, weight, and educational attainment). Given our large sample size, this indicates that if there is any effect, it is negligible. Similarly, the interaction between intake and age is weakly significant for the regressions excluding abstainers only, indicating that it is also negligible if any effect exists. None of the interaction terms are significant at the 0.1% level. Consequently, we excluded the interaction terms from the analyses of local IDPs.”

- To visualize the size of the effect for various levels of daily alcohol units, we bin participants by the number of daily alcohol units consumed and show the corresponding average volume of gray and white matter for each group after controlling for standard control variables (keeping the residual of the regression). In each bar plot, we show values for males, females, and the entire sample (Figure 4, below). We used the same approach for regional gray matter volume IDPs (Extended data Figure 2). We also tested for significant differences in GMV between each bin to the group drinking up to one unit per day, for each sex and overall. We find results that are comparable across sexes.

Figure 3. Bar plots representing the average volume of whole-brain gray and white matter volume for individuals grouped by the number of daily alcohol units after controlling for standard control variables (keeping the regression residual). The mean residuals are in terms of standard deviations of the dependent variable. The error bars represent the 95% confidence interval. * $p < 0.01$ and ** $p < .0001$ for groups showing a significant difference against the group consuming up to one alcohol unit daily.

2. The literature review on alcohol effects on the brain is sparse and could be more informative.

The introduction of the revised manuscript has been revised to include a more comprehensive literature review.

3. It appears that heavy drinking was defined as 4+ drinks/day. This cut-point, of course, would include a large percentage of people with alcohol use disorder. The authors note that their results are “predominantly driven by heavy drinkers” but “effects were also observed among individuals who reported consuming two units/day of alcohol.” Was the two units/day effect significant on its own or only when included in the regression analysis?

Thank you for this comment. When revising the manuscript, we were able to roughly double our sample size to 36,678 by incorporating a new release of brain images from the UKB. The increased sample size provided greater statistical power to identify more fine-grained effects. For example, our results now show that going from one to two units of alcohol per day is associated with the loss of gray and white matter, with an effect size that is equivalent to age-related degeneration from ages 50 to 52 (see Tables 3A and 3B below). In addition, we tested for a difference in global GMV/WMV (see Figure 3 above) and regional GMV IDPs (Extended Data Figure 2) between subjects drinking less than one unit a day and subjects drinking 1 to 2 units per day and found a large number of significant differences. All indicate that the effect is significant on its own.

The relevant text from the manuscript is provided below.

Table 3A. Predicted average additional effect (in standard deviations of IDP) of increasing alcohol intake by one daily unit on whole-brain gray matter volume and white matter volume, for models with different sets of controls (first and second columns), and for standard controls with samples excluding abstainers (third column) and heavy drinkers (last column).

Intake changes	Standard controls		Extended controls		Excluding abstainers		Excluding heavy drinkers	
	Global GMV	Global WMV	Global GMV	Global WMV	Global GMV	Global WMV	Global GMV	Global WMV
0 to 1 unit	-0.030	-0.020	-0.038	-0.017	-0.019	-0.015	-0.034	-0.019
1 to 2 units	-0.127	-0.074	-0.126	-0.073	-0.123	-0.070	-0.107	-0.067
2 to 3 units	-0.223	-0.129	-0.214	-0.129	-0.226	-0.124	-0.181	-0.116
3 to 4 units	-0.319	-0.184	-0.302	-0.185	-0.330	-0.179	-0.255	-0.164
0 to 4 units	-0.699	-0.407	-0.682	-0.404	-0.699	-0.388	-0.577	-0.367

Note. GMV = gray matter volume; WMV = white matter volume.

Table 3B. Equivalent effect of aging in terms of additional years for an average 50-year old individual.

Intake changes	Standard controls			
	Global GMV	Equivalent aging at 50	Global WMV	Equivalent aging at 50
0 to 1 unit	-0.030	0.5 years	-0.020	0.5 years
1 to 2 units	-0.127	2.0 years	-0.074	2.0 years
2 to 3 units	-0.223	3.5 years	-0.129	3.5 years
3 to 4 units	-0.319	4.9 years	-0.184	4.9 years
0 to 4 units	-0.699	10.2 years	-0.407	10.4 years

Note. GMV = gray matter volume.

“Figure 3 displays the averages of the two global IDPs in sub-samples binned according to their daily alcohol consumption range, illustrating the non-linear nature of the relationship between daily units and the global measures. The figure includes statistical tests that compare the average of the IDPs in the different sub-samples to their average in participants who consume one daily unit or less. These tests identify statistically significant effects for all bins of participants consuming more than one daily units, including those consuming as little as 1-2

daily units. These effects are observed both in the full sample and within sex.”

4. What are the error bars in the figures – standard errors or standard deviations?

The error bars are 95% confidence intervals. We added this information to the figure captions.

5. The data are presented as regression residuals so the reader has no notion of the magnitude of the effects. The authors note that the effects they found were “small but significant.” This is the strength and weakness of large sample sizes – highly significant but small effects are revealed. How meaningful were the "significant" effects? How do those effects comport with the extensive (but not thoroughly referenced) literature on the effects of alcohol use disorder on the brain? How do they comport with other epidemiological studies that considered a wide range of drinking?

We agree that it is very important to report the effect size in large samples, rather than statistical significance alone. We also agree that benchmarking our effects against other effects in the literature could help the reader gauge their size.

To illustrate the size of this effect, we added Table 3A to the manuscript (see above), which shows the predicted effect of increasing consumption by one alcohol unit daily. You are correct that these effects are in terms of regression residuals – but please note that these residuals represent interpretable units: standard deviations of IDPs (as the dependent variable in our regression was standardized).

To further increase the interpretability of the effect sizes, Table 3B includes the equivalent effects in GMV and WMV changes due to aging, for an average 50-year-old UKB participant (far-right column). For example, the effect of increasing consumption from 2 to 3 daily drinks is equivalent to the effect of aging 3.5 years for an average 50-year-old. We believe that this benchmark makes the size of these effects more interpretable and, therefore, added it to the abstract.

Finally, we shifted the focus of our discussion of the manuscript results from statistical significance to the size of the effects in terms of the variance in the IDPs explained by alcohol intake (above the control variables). We provide the relevant paragraphs below.

“Alcohol intake explains 1% of the variance in global GMV and 0.3% of the variance in global WMV across individuals beyond all other control variables (both $p < 10^{-16}$). Additional analyses excluding abstainers ($N = 33,733$) or heavy drinkers ($N = 34,383$), as well as models using an extended set of covariates (addition of BMI, educational attainment, and weight; $N = 36,678$) yield similar findings, though the variance explained by alcohol intake beyond other control variables is reduced to 0.4% for GMV and 0.1% for WMV when heavy drinkers are excluded (Extended Data Tables 1 and 2).”

“We observe the strongest effects in frontal, parietal, and insular cortices, temporal and cingulate regions, putamen, amygdala and the brain stem. In these regions, alcohol intake explains between 0.3%-0.4% of the variance in local GMV above the other covariates. Extended Data Figure 1 illustrates the marginal effect of increasing daily alcohol units on regional GMV IDPs, grouped by lobe. All of the associations are negative, except the association involving the right pallidum—where the effect size is positive but very small ($\Delta R^2 < 0.0005$). Importantly, the largest regional effect was less than half the size of the association between drinking and global GMV, indicating that the global reduction in GMV associated with alcohol intake is the result of

aggregating smaller effects that are widespread across the brain (rather than constrained to specific areas)."

"Twenty-two WM tract regions show the most consistent associations with lower FA and higher ISOVF and MD. The strongest effects of these are in the fornix, where WM integrity was previously found to be affected by drinking in studies of populations with AUD^{3,21,23}. In the fornix, alcohol intake accounts for 0.45% of the variance in ISOVF, 0.35% of the variance in MD, and 0.32% of the variance in FA. Other WM tract regions showing a similar pattern yet with effects of weaker magnitude include commissural fibers (genu and body of the corpus callosum, bilateral tapetum), projection fibers (bilateral anterior corona radiata), associative fibers (fornix cres+stria terminalis, left inferior longitudinal fasciculus), and the bilateral anterior thalamic radiations.

Among the NODDI measures, ISOVF showed the strongest effects of alcohol intake all over the brain, most notably in the tract regions discussed above. The associations between drinking and ICVF are also consistently negative yet smaller in size, with daily alcohol intake explaining no more than 0.1% of the variance in all ICVF IDPs. The associations with OD, which is a measure of tract complexity, are either positive, negative or absent, and while some are statistically significant, they are all very small in size ($\Delta R^2 < 0.001$ for all IDPs)."

6. The authors contrast their approach, which sought linear to nonlinear relations between alcohol consumption variables and brain macrostructural and microstructural integrity, with other published reports based on linear testing only. Yet, the findings in the figures do not show any indication of relations other than linear. Can the authors provide an explanation for the difference, or be clearer on noting where complex functions best describe relations?

Following your suggestions, we ran several additional analyses.

First, we added scatter plots with local polynomial regression lines (LOWESS) regression lines for total gray and white matter volumes. These regressions (see Figures 1 and 2 above) show slight concavity in the distributions. We, therefore, included both linear and quadratic values for alcohol intake and age in all of our regressions:

$$IDP_i = \beta_0 + \beta_1 X_i + \beta_2 X_i^2 + \beta_3 X_i \times SEX_i + \beta_4 X_i \times AGE_i + \gamma Z_i + e_i,$$

where IDP_i is the IDP normalized for head size, X_i is the standardized alcohol intake in $\log(1 + \text{daily units})$, AGE_i is the standardized age, Z_i is a vector of control variables, and e_i is the error term. This information was added to the Methods section. The quadratic term is highly significant in all of our whole-brain IDPs regressions ($p < 0.00001$). Please also note that our main exploratory variable – daily alcohol intake – is logged.

To demonstrate the nature of the effects, we added Tables 3A and 3B (above), which present the effects associated with increasing alcohol intake by 1 unit across the drinking spectrum and benchmark these effects to the effects of aging. We hope that these tables help the reader gauge the relationship between alcohol intake and global IDPs.

7. Please provide scatterplots with regression functions by sex of the significant brain regions over the drinking variable and over age--distinguishing men and women by color, e.g., red for women and blue for men. To be meaningful to the reader scatterplots of the brain values by drinking variable (which should not be quantized but be continuous) should show confidence interval (should be small given the large number of subjects) plus 1 and 2 standard deviations to demonstrate the variance.

Thank you for this great suggestion. The scatter plots (Figures 1 and 2, above) were highly informative and are now included in the manuscript.

8. The authors note a study limitation that the self-reported alcohol consumption covers only the past year, which would not account for people with past diagnosis of AUD. This shortcoming needs to be addressed, given that while the brain shows some recovery with prolonged sobriety, it can vary with age and sex and in fact may not recover completely.

We agree and have added this information to the discussion/limitations section.

“An additional limitation stems from the self-reported alcohol intake measures in the UK Biobank, which cover only the past year. Such estimates do not adequately reflect drinking prior to the past year and are susceptible to reporting and recall bias^{38,39}. Further, our analyses do not account for individuals with a past diagnosis of AUD. Earlier studies have shown that the brain shows some recovery with prolonged sobriety, but this recovery varies with age and sex, and recovery might be incomplete⁵⁸⁻⁶⁰. Thus, a past diagnosis of AUD would likely influence our results. We hope that future studies will shed light on how a history of AUD with prolonged recovery influences brain structure in middle-aged and older adults.”

9. The following two sentences appear to be in conflict--the last sentence of the Results and third sentence of the Discussion (page 14). Please clarify.

Thank you for identifying this conflict. We removed these sentences from the manuscript and clarified the results discussion.

10. Can the authors provide a statement that relates to specific brain networks that appear to be most affected by drinking heavily? Are they the same in men and women and over age?

We appreciate this suggestion and have expanded our results and discussion of the gray and white matter systems affected by alcohol intake. We provide the relevant paragraphs below. Please see our response to your comment #1 concerning interactions with sex and age.

“Relationship between regional GMV and alcohol intake. *To investigate whether the reduction in global GMV associated with alcohol intake stems effects of drinking in specific regions, we estimate regression models to quantify the association of alcohol intake with a total of 139 regional GMV IDPs. These IDPs were derived using parcellations from the Harvard-Oxford cortical and subcortical atlases and Diedrichsen cerebellar atlas. Of the 139 GMV IDPs, 125 (88.9%) are significantly associated with log alcohol intake (see Extended Data Table 3). We observe the strongest effects in frontal, parietal, and insular cortices, temporal and cingulate regions, putamen, amygdala and the brain stem. In these regions, alcohol intake explains between 0.3%-0.4% of the variance in local GMV above the other covariates. Extended Data Figure 1 illustrates the marginal effect of increasing daily alcohol units on regional GMV IDPs, grouped by lobe. All of the associations are negative, except the association involving the right pallidum—where the effect size is positive but very small ($\Delta R^2 < 0.0005$). Importantly, the largest regional effect was less than half the size of the association between drinking and global GMV, indicating that the global reduction in GMV associated with alcohol intake is the result of aggregating smaller effects that are widespread across the brain (rather than constrained to specific areas).*

In a similar fashion to the analysis using the global IDPs, we calculate the average localized GMV IDP for each daily alcohol unit bin (Extended Data Figure 2) and test their

difference against the average of the group drinking up to one unit per day, within sexes and in the overall sample. As expected, the number of regional GMV IDPs showing a significant negative association with alcohol intake, as well as these associations' magnitudes, increases as the average number of daily alcohol units increases. There are few regions where lower GMV is either not observed as a function of drinking (e.g., pallidum) or only apparent among heavy drinkers (e.g., fusiform cortex). However, in most regions, GMV reduction is already visible in the groups that drink moderately (i.e., consuming 1-2 or 2-3 daily units). Thus, the influence of moderate alcohol intake on GMV also appears to be widespread across the brain, and it is detectable in both males and females.

Relationship between regional WM microstructure and alcohol intake. To evaluate how drinking influence the different indicators of WM integrity at the regional level, we estimate linear regressions to quantify the association of alcohol intake with 375 IDPs, including FA, MD, ICVF, ISOVF and OD measures extracted via averaging parameters across 74 WM tract regions⁴⁵. Of the 375 WM microstructure IDPs, 179 (47.7%) are significantly associated with alcohol intake (Extended Data Table 4). Generally, alcohol intake is related to lower coherence of water diffusion, lower neurite density, and higher magnitude of water diffusion, indicating less healthy WM microstructure with increasing alcohol intake.

To visualize the magnitude of WM microstructure IDP associations with alcohol intake, Figure 4 displays the statistically significant and non-significant effects, alongside the average change in normalized WM microstructure IDPs associated with mean daily alcohol intake increasing from 2 to 3 units. Twenty-two WM tract regions show the most consistent associations with lower FA and higher ISOVF and MD. The strongest effects of these are in the fornix, where WM integrity was previously found to be affected by drinking in studies of populations with AUD^{3,21,23}. In the fornix, alcohol intake accounts for 0.45% of the variance in ISOVF, 0.35% of the variance in MD, and 0.32% of the variance in FA. Other WM tract regions showing a similar pattern yet with effects of weaker magnitude include commissural fibers (genu and body of the corpus callosum, bilateral tapetum), projection fibers (bilateral anterior corona radiata), associative fibers (fornix cres+stria terminalis, left inferior longitudinal fasciculus), and the bilateral anterior thalamic radiations.

Among the NODDI measures, ISOVF showed the strongest effects of alcohol intake all over the brain, most notably in the tract regions discussed above. The associations between drinking and ICVF are also consistently negative yet smaller in size, with daily alcohol intake explaining no more than 0.1% of the variance in all ICVF IDPs. The associations with OD, which is a measure of tract complexity, are either positive, negative or absent, and while some are statistically significant, they are all very small in size ($\Delta R^2 < 0.001$ for all IDPs)."

11. There is no discussion about the brain regions affected with regard to their representing brain systems likely affected by heavy drinking. In other words, the authors seek little in the way of neurobiological explanations of their findings.

In response to this suggestion, we revised the results and discussion sections to include additional interpretations of the effects. The relevant discussion paragraphs are provided below.

"Although nearly 90% of all regional GMVs show significant negative associations with alcohol intake, the most extensively affected regions included the frontal, parietal, and insular cortices, with deficits also in temporal and cingulate regions. Associations are also marked in the brain stem, putamen, and amygdala. The share of variance explained by alcohol intake for these regions is smaller in size than for global GMV, suggesting that the latter is the result of aggregation of many small effects that are widespread, rather than a localized effect that is limited to specific regions. Alcohol intake is further associated with 'poorer' white matter microstructure (lower FA and higher ISOVF and MD) in specific classes of white matter tracts. The commissural fibers (genu and body of the corpus callosum, bilateral tapetum), projection

fibers (bilateral anterior corona radiata), associative bundles (fornix, fornix cres+stria terminalis, left inferior longitudinal fasciculus), and the bilateral anterior thalamic radiations show the most consistent associations with alcohol intake, with the fornix showing the strongest effects. The fornix is the primary outgoing pathway from the hippocampus⁵⁰, and white matter microstructural alterations in the fornix are consistently associated with heavy alcohol use and memory impairments^{3,51}. Moreover, recent research indicates that one extreme-drinking episode can cause acute white matter damage to the fornix, suggesting that the fornix may be particularly vulnerable to alcohol's effects.

Our findings are partly consistent with studies of individuals with AUD^{18,52}. The pattern of microstructural alterations in our general population sample show that widespread WM alterations are present across multiple white matter systems. Like individuals with AUD, alcohol intake in this healthy population sample is associated with microstructural changes in superficial WM systems functionally related to GM networks, including the frontoparietal control and attention networks, and the default mode, sensorimotor, and cerebellar networks. Deeper WM systems (superior longitudinal fasciculus and dorsal frontoparietal systems, inferior longitudinal fasciculus system, and deep frontal white matter) thought to be involved in cognitive functioning by regulating reciprocal connectivity^{52,53} are also associated with alcohol intake. Within these WM systems, alcohol intake is most strongly associated with ISOVF, MD, and FA WM microstructure indices; whereas, associations with ICVF are small, and OD associations are inconsistent or nonexistent. Alcohol intake shows positive associations with ISOVF and MD and negative associations with FA. This pattern of alcohol-associated WM microstructural disruption supports previous research showing excessive intracellular and extracellular fluid in individuals with AUD²⁰. Given that alcohol increases blood-brain permeability⁵⁴ and activates pro-inflammatory cytokines in the brain⁵⁵, the association between alcohol intake and higher ISOVF (extracellular water diffusion) may be due to inflammatory demyelination. For example, higher ISOVF is evident in WM lesions of multiple sclerosis, characterized histopathologically by inflammatory demyelination associated with blood-brain permeability and axonal injury^{56,57}. Additional research is warranted; however, these findings suggest that even low-moderate alcohol intake increases intracellular and extracellular water diffusion in WM, which may be a result of alcohol-induced inflammatory demyelination.”

12. Further, they make no mention of having identified brain pathology despite the likelihood of having a large representation of Alcohol Use Disorder in the cohort.

Please note that we now observe effects in the subsample of participants who reported drinking 1-2 units daily (see our response to your comment #3).

We also added discussion of the possibility that our sample includes some individuals with a past diagnosis of AUD as a limitation (below).

“An additional limitation stems from the self-reported alcohol intake measures in the UK Biobank, which cover only the past year. Such estimates do not adequately reflect drinking prior to the past year and are susceptible to reporting and recall bias^{38,39}. Further, our analyses do not account for individuals with a past diagnosis of AUD. Earlier studies have shown that the brain shows some recovery with prolonged sobriety, but this recovery varies with age and sex, and recovery might be incomplete⁵⁸⁻⁶⁰. Thus, a past diagnosis of AUD would likely influence our results. We hope that future studies will shed light on how a history of AUD with prolonged recovery influences brain structure in middle-aged and older adults.”

Reviewer 2:

1. The general problem with the manuscript is that there is a lack of alternative hypotheses, analytical interpretation, or dimensionality in the current results. The reader is given essentially a very descriptive set of results, with no counterpoints. Non-significant results are important, but unreported or not visualized in the paper. All 7 measures are tested, but if there are different results across measures, these differences aren't tested statistically, and the authors make no conclusions about these different patterns across different white matter metrics. My suggestions in this respect would be to "dive deeper" in at least 3 different ways: (1) test for explicit regional differences (e.g., sensory vs. association regions, early vs. late myelinating regions), (2) explicit tests for the utility of different myelin metrics in detecting alcohol-related changes in the brain (are effects greater for ICVF than MD, reflecting greater changes in fiber organization vs overall diffusivity?), or (3) more explicit interaction tests for some of the many control variables interact (e.g., how does age interact with alcohol use in determining white matter health?, or are declines in WM with increasing alcohol consumption steeper in different socioeconomic groups?). The Biobank is a rich dataset, many interesting questions are possible!

To make it clear that we did not present results selectively, we added the following statement to the methods section:

"All results are available to the readers in extended data figures and tables. Specifically, Extended Data Tables 3 and 4 include the regression coefficients, p-values and incremental variance explained above that of control variables for all of the regional IDPs (both significant and non-significant). Extended Data Figure 2 includes the average GMV of all regions tested (both significant and non-significant), in bins of participants with different daily alcohol intake levels."

Further, in this revision, we roughly doubled our sample size and carried out several additional analyses to address your comments. We also made efforts to highlight the goal of our study in the final paragraph of the introduction:

"Our sample size provides us statistical power of 90% to detect effects as small as $f^2 > 0.00078$ at the 5% significance level, after accounting for multiple hypotheses testing ($p_{\text{uncorrected}} < 1.64 \times 10^{-4}$). Given previous findings, we hypothesized to see a reduction in global GMV and WMV in heavy drinkers. However, the large general population sample provided sufficient sensitivity to qualitatively and quantitatively assess how effects vary across the entire drinking spectrum and test at what threshold effects emerge. Our well-powered design also allowed us to explore whether the effects of alcohol intake on GMV and WM microstructure are localized in specific regions or conversely widespread across the brain and compare the effects across various WM integrity indicators."

Given the large sample size, we shifted the focus of our results, reporting from statistical significance to the size of the effects. For the global IDPs, we also provide, in Tables 3A, estimates of the effects resulting from increasing daily alcohol intake by one unit across the range of drinking. In Table 3B, we also benchmark these effects to those of aging for an average 50-year-old UK Biobank participant. The relevant paragraph discussing these results and the Tables are shown below for convenience.

"We estimate linear regressions to quantify the relationships between daily alcohol intake, as well as its interactions with age and sex, and the global IDPs. Our main analyses ($N = 36,585$)

controls for age, height, handedness, sex, smoking status, socioeconomic status, genetic ancestry, and county of residence (see Methods). Table 2 summarizes the results, revealing that both global IDPs decrease as a function of daily alcohol intake. Alcohol intake explains 1% of the variance in global GMV and 0.3% of the variance in global WMV across individuals beyond all other control variables (both $p < 10^{-16}$). Additional analyses excluding abstainers ($N = 33,733$) or heavy drinkers ($N = 34,383$), as well as models using an extended set of covariates (addition of BMI, educational attainment, and weight; $N = 36,678$) yield similar findings, though the variance explained by alcohol intake beyond other control variables is reduced to 0.4% for GMV and 0.1% for WMV when heavy drinkers are excluded (Extended Data Tables 1 and 2).”

Table 3A. Predicted average additional effect (in standard deviations of IDP) of increasing alcohol intake by one daily unit on whole-brain gray matter volume and white matter volume, for models with different sets of controls (first and second columns), and for standard controls with samples excluding abstainers (third column) and heavy drinkers (last column).

Intake changes	Standard controls		Extended controls		Excluding abstainers		Excluding heavy drinkers	
	Global GMV	Global WMV	Global GMV	Global WMV	Global GMV	Global WMV	Global GMV	Global WMV
0 to 1 unit	-0.030	-0.020	-0.038	-0.017	-0.019	-0.015	-0.034	-0.019
1 to 2 units	-0.127	-0.074	-0.126	-0.073	-0.123	-0.070	-0.107	-0.067
2 to 3 units	-0.223	-0.129	-0.214	-0.129	-0.226	-0.124	-0.181	-0.116
3 to 4 units	-0.319	-0.184	-0.302	-0.185	-0.330	-0.179	-0.255	-0.164
0 to 4 units	-0.699	-0.407	-0.682	-0.404	-0.699	-0.388	-0.577	-0.367

Note. GMV = gray matter volume; WMV = white matter volume.

Table 3B. Equivalent effect of aging in terms of additional years for an average 50-year old individual.

Intake changes	Standard controls			
	Global GMV	Equivalent aging at 50	Global WMV	Equivalent aging at 50
0 to 1 unit	-0.030	0.5 years	-0.020	0.5 years
1 to 2 units	-0.127	2.0 years	-0.074	2.0 years
2 to 3 units	-0.223	3.5 years	-0.129	3.5 years
3 to 4 units	-0.319	4.9 years	-0.184	4.9 years
0 to 4 units	-0.699	10.2 years	-0.407	10.4 years

Note. GMV = gray matter volume.

We examined the interactions between sex and alcohol intake and between age and alcohol intake in the regression analysis of global GMV and WMV. These interactions were very weak, especially when considering the ample statistical power of our study. Therefore, we did not include these interaction terms in the analyses of localized GMV and WM integrity measures that follow. The relevant text from the results section is below.

“In the eight regressions we tested, the interaction between alcohol intake and sex is not significant at the 1% level, except weakly for GMV when including the extended control variables (BMI, weight, and educational attainment). Given our large sample size, this indicates that if there is any effect, it is negligible. Similarly, the interaction between intake and age is weakly significant for the regressions excluding abstainers only, indicating that it is also negligible if any effect exists. Consequently, we excluded the interaction terms from the analyses of local IDPs.”

To provide context for the regional GMV results, we explicitly discuss the percent of variance explained in these IDPs above the control variables and compare it to the effects of global GMV. In general, local effects appear to be widespread across the brain and smaller than global ones (the largest local effect is half the size of the global effect in terms of variance explained). Based on these results, we revised the relevant paragraph in the results section accordingly:

“To investigate whether the reduction in global GMV associated with alcohol intake stems effects of drinking in specific regions, we estimate regression models to quantify the association of alcohol intake with a total of 139 regional GMV IDPs. These IDPs were derived using parcellations from the Harvard-Oxford cortical and subcortical atlases and Diedrichsen cerebellar atlas. Of the 139 GMV IDPs, 125 (88.9%) are significantly associated with log alcohol intake (see Extended Data Table 3). We observe the strongest effects in frontal, parietal, and insular cortices, temporal and cingulate regions, putamen, amygdala and the brain stem. In these regions, alcohol intake explains between 0.3%-0.4% of the variance in local GMV above the other covariates. Extended Data Figure 1 illustrates the marginal effect of increasing daily alcohol units on regional GMV IDPs, grouped by lobe. All of the associations are negative, except the association involving the right pallidum—where the effect size is positive but very small ($\Delta R^2 < 0.0005$). Importantly, the largest regional effect was less than half the size of the association between drinking and global GMV, indicating that the global reduction in GMV associated with alcohol intake is the result of aggregating smaller effects that are widespread across the brain (rather than constrained to specific areas).”

We explore which of the different white matter microstructural metrics best explains alcohol-related changes in the brain and where these differences occur. Given the large sample, this analysis also focuses on the size of the effect (in terms of variance in the IDP explained by alcohol intake) rather than statistical significance. The relevant paragraphs from the manuscript are shown below for convenience.

“To evaluate how drinking influence the different indicators of WM integrity at the regional level, we estimate linear regressions to quantify the association of alcohol intake with 375 IDPs, including FA, MD, ICVF, ISOVF and OD measures extracted via averaging parameters across 74 WM tract regions⁴⁵. Of the 375 WM microstructure IDPs, 179 (47.7%) are significantly associated with alcohol intake (Extended Data Table 4). Generally, alcohol intake is related to lower coherence of water diffusion, lower neurite density, and higher magnitude of water diffusion, indicating less healthy WM microstructure with increasing alcohol intake.

To visualize the magnitude of WM microstructure IDP associations with alcohol intake, Figure 4 displays the statistically significant and non-significant effects, alongside the average change in normalized WM microstructure IDPs associated with mean daily alcohol intake increasing from 2 to 3 units. Twenty-two WM tract regions show the most consistent

associations with lower FA and higher ISOVF and MD. The strongest effects of these are in the fornix, where WM integrity was previously found to be affected by drinking in studies of populations with AUD^{3,21,23}. In the fornix, alcohol intake accounts for 0.45% of the variance in ISOVF, 0.35% of the variance in MD, and 0.32% of the variance in FA. Other WM tract regions showing a similar pattern yet with effects of weaker magnitude include commissural fibers (genu and body of the corpus callosum, bilateral tapetum), projection fibers (bilateral anterior corona radiata), associative fibers (fornix cres+stria terminalis, left inferior longitudinal fasciculus), and the bilateral anterior thalamic radiations.

Among the NODDI measures, ISOVF showed the strongest effects of alcohol intake all over the brain, most notably in the tract regions discussed above. The associations between drinking and ICVF are also consistently negative yet smaller in size, with daily alcohol intake explaining no more than 0.1% of the variance in all ICVF IDPs. The associations with OD, which is a measure of tract complexity, are either positive, negative or absent, and while some are statistically significant, they are all very small in size ($\Delta R^2 < 0.001$ for all IDPs)."

2. A related, but more minor concern, is that the authors present a wealth of graphs/data, but it is poorly contextualized. I found myself searching through many different Supplementary documents to look for trends in the results, because these were not spelled out explicitly in the text. The figures in the main document aren't especially compelling, or visually easy to localize to specific brain regions. Perhaps the authors could reuse the images in Fig1 to help with this.

Thank you for this important comment. We changed all of the figures and tables based on our new analyses and findings, making efforts to demonstrate the results more intuitively. We hope that the revised manuscript is easier to navigate.

3. The authors regularly refer to "tracts", but these are more properly regions-of-interest (ROIs) within white matter regions—no tractography was performed in this study, and the inferences of canonical fiber systems are based on volumetric ROIs. As such, referring to "tracts" is misleading.

We appreciate this comment. In the revision, we describe the ROIs as white matter tract regions. These IDPs are obtained by diffusion-weighted imaging executed by the UK Biobank. They further indicate that "long-range estimates based on tract-tracing (tractography) reflect structural connectivity between pairs of brain regions". We hope this change adequately addresses this issue. The reader can refer to the cited UK Biobank papers for additional details:

- Smith, S., Alfaro-Almagro, F. & Miller, K. UK Biobank brain imaging documentation. https://biobank.ctsu.ox.ac.uk/crystal/crystal/docs/brain_mri.pdf. (2020).
- Alfaro-Almagro, F. *et al.* Image processing and Quality Control for the first 10,000 brain imaging datasets from UK Biobank. *Neuroimage* 166, 400-424 (2018).
- Miller, K. L. *et al.* Multimodal population brain imaging in the UK Biobank prospective epidemiological study. *Nat Neurosci.* 19, 1523-1536 (2016).

4. How many individuals are in each model? The authors start with 19,825, loose 747 due to data quality, and then in each of the several models run, the number of samples "decreased when phenotype data were missing", so it's unclear how many subjects contribute to each model. How many individuals were excluded based on IDP values greater than 4 standard deviations? This seems a rather lenient threshold for inclusion.

We added the number of observations in each model to Extended Data Tables 1 and 2. We also added this information in the results section:

“We estimate linear regressions to quantify the relationships between daily alcohol intake, as well as its interactions with age and sex, and the global IDPs. Our main analyses (N = 36,585) controls for age, height, handedness, sex, smoking status, socioeconomic status, genetic ancestry, and county of residence (see Methods). Table 2 summarizes the results, revealing that both global IDPs decrease as a function of daily alcohol intake. Alcohol intake explains 1% of the variance in global GMV and 0.3% of the variance in global WMV across individuals beyond all other control variables (both $p < 10^{-16}$). Additional analyses excluding abstainers (N = 33,733) or heavy drinkers (N = 34,383), as well as models using an extended set of covariates (addition of BMI, educational attainment, and weight; N = 36,678) yield similar findings, though the variance explained by alcohol intake beyond other control variables is reduced to 0.4% for GMV and 0.1% for WMV when heavy drinkers are excluded (Extended Data Tables 1 and 2).”

We further provide a more detailed explanation of how we reached the number of individuals in our sample in the Methods section. This also includes a rationale for the exclusion of individuals based on IDP values greater than 4 standard deviations in the manuscript. We also investigated how the results change when either including these excluded participants in the analysis or when using an exclusion threshold of 3 standard deviations and found that this has no impact on our findings. The relevant text from the methods section is provided below.

“The data provided by the UK Biobank and was already subject to quality control⁶¹. We excluded individuals with IDP values outside a range of four standard deviations (SDs). We chose this lenient threshold as a non-trivial number of observations (97 for GM, 127 for WM) fall between three and four SDs away from the mean, given the large sample size. The IDPs beyond the four SD range are likely the results of processing errors, or the corresponding individuals present severe brain irregularities (5 individuals for GM, 7 for WM). Note that the exclusion of these outliers does not change the statistical significance nor the magnitude of the effects that we report. The exclusion of individuals falling within three SDs of the mean does not change the results either.”

5. What motivation do the authors have for including 3 age-related variables (linear, 2nd, and 3rd order age effects)?

After producing the scatterplots of IDPs against age, as suggested by reviewer 1 (see Figure 1 below), it appeared that the third-order age effect was unnecessary. We, therefore, removed it from the regression models. We maintained the second-order effect to account for the slight concavity observed in the LOWESS regression line.

Figure 1. Scatter plots of whole-brain standardized gray matter volume (women, upper left; men, upper right) and standardized white matter volume (women, lower left; men, lower right), all normalized for head size, against the individual's age (x-axis). The plots also show the LOWESS regression line (smoothness: $a=0.2$). The 95% confidence interval is indistinguishable from the regression line. The colors are representative of the average daily alcohol consumption.

6. There are few details about how the authors apply the Holm method to their statistical testing.

We added information about the Holm method into the manuscript.

“To control the family-wise error rate in the regional GMV and WM microstructure analysis, we determine the significance thresholds for all regressions using the Holm method⁶⁸, ensuring a family-wise error rate below 5%. When testing for M hypotheses, this method orders the corresponding p -values from lowest to highest: p_0, \dots, p_M , and identifies the minimal index k such that $p_k > 0.05 / (M+1-k)$. All hypotheses with an index $m < k$ are then considered to be statistically significant. In our application, the significance threshold was determined to be 1.64×10^{-4} .”

Reviewer 3:

1. This is an important study well-powered to examine effects of low to moderate alcohol consumption on brain structure, which has been a long-term gap in the literature. I find it compelling and it will be of broad interest to neuroimaging researchers given the recommendation that even low-moderate alcohol use impacts brain structure and should be included as a confound in structural analyses. My comments were primarily concerned with aspects of methodology and clarity of reporting the analyses and findings.

Thank you for the positive feedback. In this revision, we were able to roughly double our sample size, which further increased the statistical power. We also made significant changes to our analytic approach to address your suggestions

2. Your ‘key questions’ statements could be clearer, with careful reference to AUD versus the general population.

Based on this comment, we substantially revised the manuscript, particularly the introduction. This includes making our aims/key questions clearer and paying close attention to references to AUD versus the general population. The paragraph below from the introduction states the ‘key questions’.

“...Specifically, we assess associations between alcohol intake (i.e., mean daily alcohol units; one unit=10 ml or 8 g of ethanol) and imaging derived phenotypes (IDPs) of brain structure (total GMV, total WMV, and 139 regional GMVs), as well as 375 IDPs of WM microstructure (DTI and NODDI indices), using data from 36,678 UKB participants. Our analyses adjust for numerous covariates (see Methods for an exhaustive list of these).

Our sample size provides us statistical power of 90% to detect effects as small as $f^2 > 0.00078$ at the 5% significance level, after accounting for multiple hypotheses testing ($p_{\text{uncorrected}} < 1.64 \times 10^{-4}$). Given previous findings, we hypothesized to see a reduction in global GMV and WMV in heavy drinkers. However, the large general population sample provided sufficient sensitivity to qualitatively and quantitatively assess how effects vary across the entire drinking spectrum and test at what threshold effects emerge. Our well-powered design also allowed us to explore whether the effects of alcohol intake on GMV and WM microstructure are localized in specific regions or conversely widespread across the brain and compare the effects across various WM integrity indicators.”

3. When describing potential confounds, it would be useful to include a brief summary for each point e.g. sex (women more vulnerable than men). On this page you also state the number of WM tracts but not number of GM regions examined.

We apologize for the lack of clarity. We incorporated this suggestion and added the number of IDPs examined for regional GMV and WM microstructure.

“...Potential confounds that may be associated with individual differences in both alcohol intake and neuroanatomy include sex (women are more vulnerable than men)³⁶, body mass index (BMI) (vulnerability increases as a function of BMI)³⁷, age (older adults are more vulnerable than younger adults)^{38,39}, and genetic population structure (i.e., biological characteristics that are correlated with environmental causes)⁴⁰. Similar to other research fields, progress in this area may also be limited by publication bias⁴¹.”

“Specifically, we assess associations between alcohol intake (i.e., mean daily alcohol units; one unit=10 ml or 8 g of ethanol) and imaging derived phenotypes (IDPs) of brain structure (total GMV, total WMV, and 139 regional GMVs), as well as 375 IDPs of WM microstructure (DTI and NODDI indices), using data from 36,678 UKB participants. Our analyses adjust for numerous covariates (see Methods for an exhaustive list of these).”

4. Per supplementary tables 2-3 please clarify that Model D resulted in no significant results other than for OD - this is an interesting finding to include in the main text given OD may then be the most sensitive WM measure of more subtle changes with lower exposure to neurotoxicity.

Based on the revised statistical approach and increased sample size, all figures and tables (including supplementary materials) have been updated. Significant results are now found across metrics and are not limited to OD. Given the large sample, we now focus our discussion of the effects of alcohol on different WM metrics on the size of the effects (quantified as variance explained) rather than statistical significance. The relevant paragraphs from the results section are provided below for convenience.

Relationship between regional WM microstructure and alcohol intake. *To evaluate how drinking influence the different indicators of WM integrity at the regional level, we estimate linear regressions to quantify the association of alcohol intake with 375 IDPs, including FA, MD, ICVF, ISOVF and OD measures extracted via averaging parameters across 74 WM tract regions⁴⁵. Of the 375 WM microstructure IDPs, 179 (47.7%) are significantly associated with alcohol intake (Extended Data Table 4). Generally, alcohol intake is related to lower coherence of water diffusion, lower neurite density, and higher magnitude of water diffusion, indicating less healthy WM microstructure with increasing alcohol intake.*

To visualize the magnitude of WM microstructure IDP associations with alcohol intake, Figure 4 displays the statistically significant and non-significant effects, alongside the average change in normalized WM microstructure IDPs associated with mean daily alcohol intake increasing from 2 to 3 units. Twenty-two WM tract regions show the most consistent associations with lower FA and higher ISOVF and MD. The strongest effects of these are in the fornix, where WM integrity was previously found to be affected by drinking in studies of populations with AUD^{3,21,23}. In the fornix, alcohol intake accounts for 0.45% of the variance in ISOVF, 0.35% of the variance in MD, and 0.32% of the variance in FA. Other WM tract regions showing a similar pattern yet with effects of weaker magnitude include commissural fibers (genu and body of the corpus callosum, bilateral tapetum), projection fibers (bilateral anterior corona radiata), associative fibers (fornix cres+stria terminalis, left inferior longitudinal fasciculus), and the bilateral anterior thalamic radiations.

Among the NODDI measures, ISOVF showed the strongest effects of alcohol intake all over the brain, most notably in the tract regions discussed above. The associations between drinking and ICVF are also consistently negative yet smaller in size, with daily alcohol intake explaining no more than 0.1% of the variance in all ICVF IDPs. The associations with OD, which is a measure of tract complexity, are either positive, negative or absent, and while some are statistically significant, they are all very small in size ($\Delta R^2 < 0.001$ for all IDPs).”

5. Numerous regions were examined, and while significant regions from Model A are reported in the supplementary figures and tables, an overall statement of significant regional results should be included somewhere for context (i.e., 16/139? (%) GM regions (pg 7?) and #/27 (%) WM tracts (pg 11?) were significantly associated with alcohol intake).

We appreciate this suggestion and have incorporated the following:

“Specifically, we assess associations between alcohol intake (i.e., mean daily alcohol units; one unit=10 ml or 8 g of ethanol) and imaging derived phenotypes (IDPs) of brain structure (total GMV, total WMV, and 139 regional GMVs), as well as 375 IDPs of WM microstructure (DTI and NODDI indices), using data from 36,678 UKB participants.”

6. In the introduction you mention the key question of nonlinearity, but your analysis/reporting on that issue isn't very clear to me outside of using Models C and D and then forming the groups. Why wasn't there a direct examination of alcohol as nonlinear since the analysis is so well powered?

In response to this question, we modified the analysis in several ways to account for and confirm nonlinear relations.

(1) We first created scatter plots of IDPs against alcohol intake and added a LOWESS regression line to visualize the average effect (see Figure 2 below). This revealed a concavity in the relation.

(2) We added a quadratic term in the regression that proved highly significant (see Table 2 below).

(3) Given that the measure of alcohol intake was logged and our results include both linear and quadratic terms, we also added tables to illustrate the nature of the effects (Tables 3A and 3B, below). These tables provide the predicted changes in GMV and WMV for one unit increase relative to a baseline consumption of one to three units daily and benchmarking these effects against the effects of aging for an average 50-year-old participant in our sample.

Figure 2. Scatter plots of whole-brain standardized gray matter volume (women, upper left; men, upper right) and standardized white matter volume (women, lower left; men, lower right), all normalized for head size, against the individual's daily alcohol consumption (x-axis, in log scale). The plots also show the LOWESS regression line (smoothness: $a=0.2$), with its 95% confidence interval.

Table 2. Regression analysis with global IDPs as outcome variables. All regressions include standard controls. Intake is measured in $\log(1 + \text{daily units of alcohol})$.

Variable	Dependent variable: global GMV		Dependent variable: global WMV	
	N: 36,678 (d.f.: 36,585), R^2 : 0.514		N: 36,678 (d.f.: 36,585), R^2 : 0.514	
	Regression Coefficient (S.Err), 95% CI	t-stat (p-value)	Regression Coefficient (S.Err), 95% CI	t-stat (p-value)
intake	-0.1095 (0.0058), CI: [-0.1209,-0.0982]	-19.0 ($p < 1.0e-16$)	-0.0650 (0.0078), CI: [-0.0802,-0.0498]	-8.4 ($p < 1.0e-16$)
intake ²	-0.0651 (0.0037), CI: [-0.0723,-0.0579]	-17.7 ($p < 1.0e-16$)	-0.0370 (0.0050), CI: [-0.0468,-0.0273]	-7.5 ($p = 7.8e-14$)
intake x male	0.0174 (0.0080), CI: [0.0018,0.0330]	2.2 ($p = 2.9e-02$)	0.0164 (0.0107), CI: [-0.0046,0.0374]	1.5 ($p = 1.2e-01$)
intake x std. age	0.0080 (0.0037), CI: [0.0008,0.0152]	2.2 ($p = 3.0e-02$)	0.0111 (0.0050), CI: [0.0014,0.0208]	2.2 ($p = 2.5e-02$)
std. age	-0.5991 (0.0038), CI: [-0.6066,-0.5916]	-157.0 ($p < 1.0e-16$)	-0.3213 (0.0051), CI: [-0.3313,-0.3112]	-62.6 ($p < 1.0e-16$)

std. age ²	-0.0378 (0.0034), CI: [-0.0445,-0.0311]	-11.0 (p < 1.0e-16)	-0.0127 (0.0046), CI: [-0.0217,-0.0037]	-2.8 (p = 5.7e-03)
	Against model without intake and interactions Delta R ² : 0.0099, F-test: p < 1.0e-16		Against model without intake and interactions Delta R ² : 0.0033, F-test: p < 1.0e-16	

Table 3A. Predicted average additional effect (in standard deviations of IDP) of increasing alcohol intake by one daily unit on whole-brain gray matter volume and white matter volume, for models with different sets of controls (first and second columns), and for standard controls with samples excluding abstainers (third column) and heavy drinkers (last column).

Intake changes	Standard controls		Extended controls		Excluding abstainers		Excluding heavy drinkers	
	Global GMV	Global WMV	Global GMV	Global WMV	Global GMV	Global WMV	Global GMV	Global WMV
0 to 1 unit	-0.030	-0.020	-0.038	-0.017	-0.019	-0.015	-0.034	-0.019
1 to 2 units	-0.127	-0.074	-0.126	-0.073	-0.123	-0.070	-0.107	-0.067
2 to 3 units	-0.223	-0.129	-0.214	-0.129	-0.226	-0.124	-0.181	-0.116
3 to 4 units	-0.319	-0.184	-0.302	-0.185	-0.330	-0.179	-0.255	-0.164
0 to 4 units	-0.699	-0.407	-0.682	-0.404	-0.699	-0.388	-0.577	-0.367

Note. GMV = gray matter volume; WMV = white matter volume.

Table 3B. Equivalent effect of aging in terms of additional years for an average 50-year old individual.

Intake changes	Standard controls			
	Global GMV	Equivalent aging at 50	Global WMV	Equivalent aging at 50
0 to 1 unit	-0.030	0.5 years	-0.020	0.5 years
1 to 2 units	-0.127	2.0 years	-0.074	2.0 years
2 to 3 units	-0.223	3.5 years	-0.129	3.5 years
3 to 4 units	-0.319	4.9 years	-0.184	4.9 years
0 to 4 units	-0.699	10.2 years	-0.407	10.4 years

Note. GMV = gray matter volume.

7. Please expand your discussion of regions of GM and WM with greatest effect sizes, and whether these regions are consistent with regions demonstrated in heavier drinkers (i.e., is it that there is strong overlap and by examining low-moderate drinkers we see regions that might be most vulnerable, or is there some other pattern with low-moderate than heavy drinkers?).

We expanded our discussion of the gray matter, white matter, and network findings showing associations with alcohol intake. The relevant paragraphs from the results section are provided below for convenience.

“Relationship between regional GMV and alcohol intake. To investigate whether the reduction in global GMV associated with alcohol intake stems effects of drinking in specific regions, we estimate regression models to quantify the association of alcohol intake with a total of 139 regional GMV IDPs. These IDPs were derived using parcellations from the Harvard-Oxford cortical and subcortical atlases and Diedrichsen cerebellar atlas. Of the 139 GMV IDPs, 125 (88.9%) are significantly associated with log alcohol intake (see Extended Data Table 3). We observe the strongest effects in frontal, parietal, and insular cortices, temporal and cingulate regions, putamen, amygdala and the brain stem. In these regions, alcohol intake explains between 0.3%-0.4% of the variance in local GMV above the other covariates. Extended Data Figure 1 illustrates the marginal effect of increasing daily alcohol units on regional GMV IDPs, grouped by lobe. All of the associations are negative, except the association involving the right pallidum—where the effect size is positive but very small ($\Delta R^2 < 0.0005$). Importantly, the largest regional effect was less than half the size of the association between drinking and global GMV, indicating that the global reduction in GMV associated with alcohol intake is the result of aggregating smaller effects that are widespread across the brain (rather than constrained to specific areas).

In a similar fashion to the analysis using the global IDPs, we calculate the average localized GMV IDP for each daily alcohol unit bin (Extended Data Figure 2) and test their difference against the average of the group drinking up to one unit per day, within sexes and in the overall sample. As expected, the number of regional GMV IDPs showing a significant negative association with alcohol intake, as well as these associations' magnitudes, increases as the average number of daily alcohol units increases. There are few regions where lower GMV is either not observed as a function of drinking (e.g., pallidum) or only apparent among heavy drinkers (e.g., fusiform cortex). However, in most regions, GMV reduction is already visible in the groups that drink moderately (i.e., consuming 1-2 or 2-3 daily units). Thus, the influence of moderate alcohol intake on GMV also appears to be widespread across the brain, and it is detectable in both males and females.

Relationship between regional WM microstructure and alcohol intake. To evaluate how drinking influence the different indicators of WM integrity at the regional level, we estimate linear regressions to quantify the association of alcohol intake with 375 IDPs, including FA, MD, ICVF, ISOVF and OD measures extracted via averaging parameters across 74 WM tract regions⁴⁵. Of the 375 WM microstructure IDPs, 179 (47.7%) are significantly associated with alcohol intake (Extended Data Table 4). Generally, alcohol intake is related to lower coherence of water diffusion, lower neurite density, and higher magnitude of water diffusion, indicating less healthy WM microstructure with increasing alcohol intake.

To visualize the magnitude of WM microstructure IDP associations with alcohol intake, Figure 4 displays the statistically significant and non-significant effects, alongside the average change in normalized WM microstructure IDPs associated with mean daily alcohol intake increasing from 2 to 3 units. Twenty-two WM tract regions show the most consistent associations with lower FA and higher ISOVF and MD. The strongest effects of these are in the

fornix, where WM integrity was previously found to be affected by drinking in studies of populations with AUD^{3,21,23}. In the fornix, alcohol intake accounts for 0.45% of the variance in ISOVF, 0.35% of the variance in MD, and 0.32% of the variance in FA. Other WM tract regions showing a similar pattern yet with effects of weaker magnitude include commissural fibers (genu and body of the corpus callosum, bilateral tapetum), projection fibers (bilateral anterior corona radiata), associative fibers (fornix cres+stria terminalis, left inferior longitudinal fasciculus), and the bilateral anterior thalamic radiations.

Among the NODDI measures, ISOVF showed the strongest effects of alcohol intake all over the brain, most notably in the tract regions discussed above. The associations between drinking and ICVF are also consistently negative yet smaller in size, with daily alcohol intake explaining no more than 0.1% of the variance in all ICVF IDPs. The associations with OD, which is a measure of tract complexity, are either positive, negative or absent, and while some are statistically significant, they are all very small in size ($\Delta R^2 < 0.001$ for all IDPs)."

8. Exclusions of IDPs more than 4 SDs from the mean seems very lenient considering all of the factors that influence brain structure, especially with aging. Why was 4 SD chosen rather than 3 SD? Do sample sizes or results significantly change with the additional exclusions?

Thank you for the comment, which is very similar to a comment by Reviewer 2. The revised manuscript includes a rationale for the exclusion of individuals based on IDP values greater than 4 standard deviations in the manuscript. We also investigated how the results change when either including these participants in the analysis or using an exclusion threshold of 3 standard deviations and found that neither has an impact on our findings. The relevant text from the methods section is provided below.

"The data provided by the UK Biobank and was already subject to quality control⁶¹. We excluded individuals with IDP values outside a range of four standard deviations (SDs). We chose this lenient threshold as a non-trivial number of observations (97 for GM, 127 for WM) fall between three and four SDs away from the mean, given the large sample size. The IDPs beyond the four SD range are likely the results of processing errors, or the corresponding individuals present severe brain irregularities (5 individuals for GM, 7 for WM). Note that the exclusion of these outliers does not change the statistical significance nor the magnitude of the effects that we report. The exclusion of individuals falling within three SDs of the mean does not change the results either."

9. How skewed or otherwise non-normal was alcohol intake, assuming that underlies your log transformation?

We presented a histogram of alcohol intake in Table 1, which indicates that the distribution of alcohol intake is skewed and becomes less skewed after log-transformation.

Table 1. Empirical distributions of variables.

Variable	Females	Males	All
Sample size	19,390	17,288	36,678
Abstainers	2,006	899	2,905

Age: Mean (SD)	 63.09 (7.37)	 64.42 (7.60)	 63.72 (7.51)
Daily alcohol units: Mean (SD)	 0.87 (0.91)	 1.49 (1.32)	 1.16 (1.16)
Standardized log(1+daily units): Mean (SD)	 -0.24 (0.96)	 0.27 (0.97)	 0.00 (1.00)
Total GMV (cm ³): Mean (SD)	 593.42 (48.07)	 641.67 (52.27)	 616.16 (55.58)
Total WMV (cm ³): Mean (SD)	 514.06 (47.70)	 584.53 (54.08)	 547.27 (61.80)
Head size scaling factor (greater for smaller heads): Mean (SD)	 1.37 (0.10)	 1.21 (0.09)	 1.29 (0.12)

Note. SD = standard deviation, GMV = gray matter volume, WMV = white matter volume.

REVIEWER COMMENTS

Reviewer #1 (Remarks to the Author):

The graphical results presented in Figure 3 is concerning.

Consuming groups (1,2), (2,3), (3,4) and 4+ were tested against consuming group (0,1).

It appears that the significance for group (1,2) is due to the fact that group (0,1) had residuals greater than 0. In fact, the white matter male group (1,2) appears to have a mean of 0 but are still purported to be showing a significant deleterious effect. Could one interpret the data as presented here to say that 0-to1 drink/day is salutary? The effects from (2,3) onward for gray matter volume and (3,4) onward for white matter volume appear sound and consistent with previous literature. The (1,2) vs (0,1) comparison, however, is concerning because of the nature of the statistical comparison. Is there an alternative statistical test approach, or is this a function/limitation of the regression approach? This is not a minor issue. It is the linchpin of the authors' conclusions, has profound implications for public health policy, and is one the authors need to address. The whole regression to zero and the inference about low usage is driven by the inclusion of heavy to alcoholic use disorder (AUD) levels in the model.

The alcohol consumption data presented in Figure 2, on a log scale, do not suggest a rationale for quantizing the data into the consumption groups presented in Figure 3. The acceleration of gray and white matter volume does not begin until after at least 2 units/day. For gray matter volume, for up to 3 units/day, women are above predicted, whereas men are on average always below predicted values regardless of consumption. This suggests inadequate normalization for sex effects.

The Fig. 1 color-by-use presentation fails to reveal the drinking levels. The color and dots are too small and indiscernible. It would be relevant to the analysis to test whether there is an age-alcohol use interaction for the (1,2) group as one would expect for the 4+ group. Even with the limitation of rationale for the (1,2) group, is there a brain by age interaction, given the literature on brain-age interaction alcohol use disorder?

In Fig 4, the focus on the (2,3) group for DTI results is not justified, especially given the crux of the claim that regarding the unhealthy effects of (1,2) drinks/day. The Fornix, which shows the largest DTI effects, is rife with partial voluming probably contributing to the large FA and MD effects. This is not discussed.

The term "Controls" in the tables is misleading because it implies people rather than factors. It appears equivalent to "abstainers" and "heavy drinkers" in the two right-most columns.

Reviewer #3 (Remarks to the Author):

The authors have adequately addressed my initial concerns, and the manuscript is much improved in clarity of methods and presentation of results.

Reviewer #4 (Remarks to the Author):

"Multimodal brain imaging study of 36,678 participants reveals adverse effects of moderate drinking"

Summary: This is a very thorough study that provides a comprehensive analysis of the effects the drinking spectrum on gray and white matter. The authors have considered the responses from the previous reviewers and have fulfilled most of their requests in detail. The only potential oversight from Reviewer 2 is their request to "test for explicit regional differences (e.g., sensory vs. association regions, early vs. late myelinating regions"; however, the authors provide detailed GMV region-by-region effects of the bins of daily alcohol intake in the supplementary tables.

Minor comment: In the “Relationship between regional WM microstructure and alcohol intake” section/figure, it would be useful to have the full range of alcohol consumption groups (as shown in the GMV figures) instead of only the effect from 2 to 3 units.

Reviewer 1's comments:

The graphical results presented in Figure 3 are concerning. Consuming groups (1,2), (2,3), (3,4) and 4+ were tested against consuming group (0,1). It appears that the significance for group (1,2) is due to the fact that group (0,1) had residuals greater than 0. In fact, the white matter male group (1,2) appears to have a mean of 0 but are still purported to be showing a significant deleterious effect. Could one interpret the data as presented here to say that 0-to1 drink/day is salutary? The effects from (2,3) onward for gray matter volume and (3,4) onward for white matter volume appear sound and consistent with previous literature. The (1,2) vs (0,1) comparison, however, is concerning because of the nature of the statistical comparison. Is there an alternative statistical test approach, or is this a function/limitation of the regression approach? This is not a minor issue. It is the linchpin of the authors' conclusions, has profound implications for public health policy, and is one the authors need to address. The whole regression to zero and the inference about low usage is driven by the inclusion of heavy to alcoholic use disorder (AUD) levels in the model.

We are sorry that this was not clear to the reviewer in his/her reading of our paper. The relevant information in Figure 3 is the relative differences between the groups of drinkers, not the nominal values of the residuals nor their sign.

The process of first regressing a dependent variable on control variables and then analyzing the residuals has a long tradition in statistical analysis, starting with the Frish-Waugh approach, resulting in the Frisch–Waugh–Lovell theorem (Frisch and Waugh 1933; Lovell 1963). By definition, the residuals should average to 0 in the **whole sample, not in each group**.

To illustrate this, suppose that you regress height on the age of children and take the residuals. You then group the residuals by biological sex. You should expect that the average residual for boys is above 0, and the average residual for girls is below 0. If you want to quantify the effect of biological sex on height (controlling for age), the important metric is the relative difference between the groups (boys and girls)—not the nominal value or the sign of the average residual for each group.

In summary, the fact that the residuals are not zero in each group in Figure 3 indicates that the average group residual differs from the average whole sample residual (which should be, by construction, 0), suggesting that there is a statistical link between the explanatory variable that was not included in the first regression (alcohol intake) and the dependent variable (brain morphometry). Given that the overall average of the residual has to be zero and that alcohol has a strong negative effect on those who consume 4+ daily drinks and a much weaker effect on those who consume (1,2), it is not surprising that the residuals of those who consume only (1,2) drinks are nominally positive. Importantly, though, the metric that is relevant for quantifying the effect of drinking is the difference between this group and the group that drinks less.

In response to this comment, we revised the caption of Figure 3 to clarify this issue and avoid such misunderstandings by readers:

“Figure 3. Bar plots representing the average residual volume of whole-brain gray and white matter volume for individuals grouped by the number of daily alcohol units after controlling for standard control variables. The mean residuals are in terms of standard deviations of the dependent variable, where zero represents the average residual in the full sample. The error bars represent the 95% confidence interval. * $p < 0.01$ and ** $p < .0001$ for groups showing a significant difference against the group consuming up to one alcohol unit daily.”

The alcohol consumption data presented in Figure 2, on a log scale, do not suggest a rationale for quantizing the data into the consumption groups presented in Figure 3. The acceleration of gray and white matter volume does not begin until after at least 2 units/day.

We respectfully disagree with the reviewer’s assessment based on what we assume is the simple visualization (or eyeballing) of Figure 2.

Below, we present “zoomed in” snapshots of the trend line between 1 and 2 drinks from Figure 2 (the left side of each image is 1 drink, the right is 2 drinks). These lines show a decline between 1 and 2 drinks. Please also note that the scale here is logarithmic; the slope of the linear relationship in this range would be greater than shown in the figure.

For gray matter volume, for up to 3 units/day, women are above predicted, whereas men are on average always below predicted values regardless of consumption. This suggests inadequate normalization for sex effects.

We assume that the reviewer is referring to Figure 2. The normalization in this figure was done on the entire sample, and thus it is expected that men and women will show slightly different average volumes. The conclusion that the average woman’s (man’s) volume is above (below) predicted is thus not warranted, as the zero line in the graphs is the average for the whole group (men + women). This process is equivalent to regressing brain volume on a constant for the whole group and then standardizing the residual (again for the whole group). The issue raised is similar to the first one addressed in this reply. The fact that a point or regression line is above or below 0 in this analysis is not informative. What is important is that the regression within a given group shows a positive or negative trend. Furthermore, all of the LOWESS regressions were done within males and females separately, so that the only reason to normalize the data is to facilitate interpretation.

In response to this comment, we have clarified the fact that the zero line in Figure 2 represents the average value for the whole sample (men+women):

“The dashed line represents the average standardized volume of the full sample (men and women).”

The Fig. 1 color-by-use presentation fails to reveal the drinking levels. The color and dots are too small and indiscernible.

The color-by-use was meant to enable the reader to look at the figure globally, differentiate lighter from darker areas (clusters of light or heavy drinkers, respectively), and identify areas of uniform shading, which represent more evenly distributed alcohol consumption. While we believe this information is relevant and presented accurately, we can remove the shading altogether and make all the points the same color if the editor believes it is preferable.

It would be relevant to the analysis to test whether there is an age-alcohol use interaction for the (1,2) group as one would expect for the 4+ group. Even with the limitation of rationale for the (1,2) group, is there a brain by age interaction, given the literature on brain-age interaction alcohol use disorder?

As the reviewer mentioned potential interactions in the first round, we tested for these interactions and reported the results in the previous manuscript (line 183):

*“Similarly, the interaction between intake and age is weakly significant for the regressions excluding abstainers only, indicating that it is also negligible if any effect exists. **None of the interaction terms are significant at the 0.1% level.** Consequently, we excluded the interaction terms from the analyses of local IDPs.”*

While these interactions might have been expected, they are unfortunately so weak that we can not detect them with certainty in a sample of over 30,000 participants. We can only conclude that if they exist, they are negligible.

In response to this comment, we rephrase the relevant paragraph to clarity:

“Similarly, none of the interactions between intake and age are significant at the 0.1% level. Only the regressions that exclude abstainers were significant at the 0.1% level ($p = 0.034$ for gray matter, $p = 0.0014$ for white matter), which given the large sample size suggests that the effects are negligible. Consequently, we excluded the interaction terms from the analyses of local IDPs”.

In Fig 4, the focus on the (2,3) group for DTI results is not justified, especially given the crux of the claim regarding the unhealthy effects of (1,2) drinks/day.

The claim of the unhealthy effects of drinking (1,2) drinks/day is supported by Figure 3, where we show significant effects on the whole brain at this dosage. Figure 4 is not focused on whole-brain gray/white matter volume quantified from T1 images but rather on white matter tractography quantified from DTI images. This analysis was done using regression methods, as

indicated in the manuscript. The use of regression methods was chosen in order to quantify the effect of alcohol, whereas the previous analysis used the “difference between groups approach” to test hypotheses such as “unhealthy effects are detectable starting at 1 to 2 units/day”. The asterisks in the figure indicate that the alcohol intake variables were significant in the regression, rather than significant differences between groups (all the results are available in extended data tables).

The predicted effect of going from 2 to 3 drinks/day is presented only for illustrative purposes, and one can easily recompute the predicted effect for any other difference (from 0.4 to 3.8, for instance). It is unrelated to the significance of the regression coefficients and is not meant to answer a claim of the type “that drinking 1 to 2 drinks a day has negative effects”.

In response to this comment, we revised the caption of Figure 4 to clarify that the significance shown with the asterisks is for indicating significant regression coefficients rather than reflecting significant differences between groups. We also recreated the figure to reflect the expected effects of increasing consumption from 1 to 2 drinks (which may be more policy-relevant). We still provide the effects of increasing consumption from 2 to 3 drinks as an Extended Data Figure.

The new Figure 4 and its caption are appended below for convenience.

Figure 4. Effect of daily alcohol units on white matter microstructure indices of interest across white matter tract regions. Asterisks denote statistically significant effects, $p < 1.64 \times 10^{-4}$. Colors represent the expected change in each IDP resulting from the increase in daily consumption from 1 to 2 units, based on the regression model. r = right, l = left

The Fornix, which shows the largest DTI effects, is rife with partial voluming probably contributing to the large FA and MD effects. This is not discussed.

Thank you for this comment, as you are correct. We have added the following to the discussion:

“Finally, partial volume effects (e.g., voxels containing cerebrospinal fluid (CSF)) can reduce the accuracy of tissue characterization and WM microstructural estimates. Previous research indicates that partial volume effects can bias diffusion measures toward a pattern of high diffusivity (MD) and reduced FA, particularly in intraventricular locations like the fornix^{61,62}. As such, our findings could reflect partial volume effects; however, it should be noted that the

structural data were acquired using T2-weighted FLAIR imaging, a structural technique that mitigates CSF contamination by suppressing signal from fluid (CSF)."

The term "Controls" in the tables is misleading because it implies people rather than factors. It appears equivalent to "abstainers" and "heavy drinkers" in the two right-most columns.

This is an inter-disciplinary jargon issue. We replaced "controls" with "control variables" throughout the manuscript to avoid confusion.

Reviewer #3's comments:

The authors have adequately addressed my initial concerns, and the manuscript is much improved in clarity of methods and presentation of results.

We thank the reviewer's constructive feedback throughout the review process and for her/his positive evaluation!

Reviewer #4's comments:

This is a very thorough study that provides a comprehensive analysis of the effects of the drinking spectrum on gray and white matter. The authors have considered the responses from the previous reviewers and have fulfilled most of their requests in detail. The only potential oversight from Reviewer 2 is their request to "test for explicit regional differences (e.g., sensory vs. association regions, early vs. late myelinating regions"; however, the authors provide detailed GMV region-by-region effects of the bins of daily alcohol intake in the supplementary tables.

We thank the reviewer for the constructive feedback throughout the review process and for the positive evaluation.

Minor comment: In the "Relationship between regional WM microstructure and alcohol intake" section/figure, it would be useful to have the full range of alcohol consumption groups (as shown in the GMV figures) instead of only the effect from 2 to 3 units.

Thank you for this suggestion. We agree that this information is important, though its visualization is challenging given the large number of outcomes and because the effects include both linear and quadratic terms. In response to this comment (and a related comment by reviewer 1), we added to the main text (Figure 4) an additional visualization of the effects of increasing from 1 to 2 units. We now also still include a visualization of effects of going from 2 to 3 drinks in Extended Data Figure 3.

References

Frisch, Ragnar, and Frederick V. Waugh. 1933. "Partial Time Regressions as Compared with Individual Trends." *Econometrica: Journal of the Econometric Society* 1 (4): 387–401.

Lovell, Michael C. 1963. "Seasonal Adjustment of Economic Time Series and Multiple Regression Analysis." *Journal of the American Statistical Association* 58 (304): 993–1010.

REVIEWER COMMENTS

Reviewer #1 (Remarks to the Author):

None.

Reviewer #3 (Remarks to the Author):

The authors have sufficiently responded to remaining concerns of reviewers. I appreciate the addition of discussion of potential partial voluming effects, and the updates to Figure 4.

Minor comment that could be addressed at proofing: One of the age scales in Figure 1 does not match the others (needs additional markers).

Reviewer #4 (Remarks to the Author):

The authors have addressed all remaining concerns. Excellent paper. Thank you for the opportunity to review.

Multimodal brain imaging study of 36,678 participants reveals adverse effects of moderate drinking

Responses to reviewers

Reviewer #1

None.

None.

Reviewer #3

The authors have sufficiently responded to remaining concerns of reviewers. I appreciate the addition of discussion of potential partial voluming effects, and the updates to Figure 4.

Minor comment that could be addressed at proofing: One of the age scales in Figure 1 does not match the others (needs additional markers).

Thank you for your positive feedback! We greatly appreciate the time and effort you dedicated to our paper. We have fixed the age scale in the top left plot of Figure 1.

Reviewer #4 (Remarks to the Author):

The authors have addressed all remaining concerns. Excellent paper. Thank you for the opportunity to review.

Thank you for your positive feedback. We greatly appreciate the time that you spent on the review of our paper.

REVIEWER COMMENTS

Reviewer #5 (Remarks to the Author):

This paper uses a very large UK Biobank sample of some 35K general population individuals of middle age to show that alcohol consumption is associated with various deviations in brain structure. This alone makes the study of considerable significance.

The authors analyze the data with and without abstainers and heavy drinkers, and note that effect sizes shrink when heavy drinkers are excluded, but the effects remain. These findings are important because they show that the findings are not dependent extreme drinking behavior.

I see three problems with the paper as it currently stands. The first is that the authors are arguing (e.g., in the Abstract) that the study shows that modest to moderate drinking has a significant effect on brain health. However, it appears that the effects they are describing, such as those attributable to the number of drinks consumed a week, are based on analysis of the entire sample with the heavy drinkers included. There is nothing fundamentally wrong with including the heavy drinkers, what is wrong is then claiming that the observed effects of the magnitude described (in the Abstract, 2 beers are said to cause a brain deficit) are attributable to modest to moderate drinking. This is fixable by not making this claim, or clearly showing what the unit consumption effects are when heavy drinkers are excluded.

A second problem is the authors are essentially arguing that alcohol is causing these effects. This is a cross sectional observational study. It is not possible to make causal inferences with his design. There is substantial evidence that those at risk for developing alcohol abuse show brain anomalies before they begin drinking, and the authors of this study cannot be certain the effects they are finding are not reflective of reverse causality or some unmeasured third variable effect. This could be handled by adding consideration of these possibilities in the limitations.

Third, the observed brain deviations are not directly associated with a brain health problem. However reasonable it is to infer the existence of deficits or adverse effects as noted in the manuscript title, it is not clear from this study that there is a functional problem associated with the structural findings.

Reviewer #5 (Remarks to the Author):

This paper uses a very large UK Biobank sample of some 35K general population individuals of middle age to show that alcohol consumption is associated with various deviations in brain structure. This alone makes the study of considerable significance. The authors analyze the data with and without abstainers and heavy drinkers, and note that effect sizes shrink when heavy drinkers are excluded, but the effects remain. These findings are important because they show that the findings are not dependent on extreme drinking behavior.

Thank you for your supportive feedback!

I see three problems with the paper as it currently stands.

The first is that the authors are arguing (e.g., in the Abstract) that the study shows that modest to moderate drinking has a significant effect on brain health. However, it appears that the effects they are describing, such as those attributable to the number of drinks consumed a week, are based on analysis of the entire sample with the heavy drinkers included. There is nothing fundamentally wrong with including the heavy drinkers, what is wrong is then claiming that the observed effects of the magnitude described (in the Abstract, 2 beers are said to cause a brain deficit) are attributable to modest to moderate drinking. This is fixable by not making this claim, or clearly showing what the unit consumption effects are when heavy drinkers are excluded.

Thank you for your comment. Our claim that modest to moderate drinking is associated with reduced global GMV and WMV is supported by analyses that compared the average GMV and WMV between participants who drink 1-2 units daily, to the average GMV of participants who drink 1 or fewer units daily. This analysis (illustrated in Figure 3, appended below for convenience) shows that people who drink between one to two units daily have significantly reduced global GMV and WMV relative to the group that drinks less than 1 unit. These effects replicate when heavy drinkers are entirely excluded from the analysis, as shown in Extended Figure 1 (also appended below for convenience).

Since all of the measures of WMV microstructure in our study represent local, rather than global effects, we decided to remove the claim that drinking 2 units already affects WMV microstructure from the last sentence of the abstract, which now reads: *“However, a daily alcohol intake of as little as one to two units – 250 to 500 ml of a 4% beer or 76 to 146 ml of a 13% wine – is associated with lower global GMV and WMV, potentially placing moderate drinkers at risk of adverse brain outcomes.”*

Following your comment, we have made additional efforts to provide the readers quantification of the global effects resulting from models that exclude heavy drinkers from the analyses. Specifically, we re-calculated the effects of TABLE 3B (which benchmarks the effects of drinking on global GMV and WMV to those of age), using the models that exclude heavy drinkers. These results (showing similar yet somewhat smaller effects) are provided in the new Table 3C, appended below for convenience.

Figure 3. Bar plots representing the average residual volume of whole-brain gray and white matter volume for individuals grouped by the number of daily alcohol units after controlling for

standard control variables. The mean residuals are in terms of standard deviations of the dependent variable, where zero represents the average residual in the full sample. The error bars represent the 95% confidence interval. * $p < 0.01$ and ** $p < .0001$ for groups showing a significant difference against the group consuming up to one alcohol unit daily. Extended Data Figure 1 replicates this Figure with the exclusion of heavy drinkers.

Extended Data Figure 1.

Table 3C. Predicted equivalent effect of aging in terms of additional years for an average 50-year-old individual (model excludes heavy drinkers).

	Standard control variables			
Intake changes	Global GMV	Equivalent aging at 50	Global WMV	Equivalent aging at 50
0 to 1 unit	-0.034	0.5 years	-0.019	0.5 years

1 to 2 units	-0.107	1.7 years	-0.067	1.8 years
2 to 3 units	-0.181	2.9 years	-0.116	3.1 years
3 to 4 units	-0.255	4.0 years	-0.164	4.4 years
0 to 4 units	-0.577	8.6 years	-0.367	9.5 years

Note. GMV = gray matter volume.

A second problem is the authors are essentially arguing that alcohol is causing these effects. This is a cross sectional observational study. It is not possible to make causal inferences with his design. There is substantial evidence that those at risk for developing alcohol abuse show brain anomalies before they begin drinking, and the authors of this study cannot be certain the effects they are finding are not reflective of reverse causality or some unmeasured third variable effect. This could be handled by adding consideration of these possibilities in the limitations

We 100% agree. Following your comment, we toned down causal claims in the manuscript. Given that the term “adverse effects” might be interpreted by readers as a claim of causality, we revised the title to: “Multimodal brain imaging study of 36,678 participants reveals lower gray and white matter volumes in moderate drinkers”.

We also added the following paragraph to the discussion of limitations:

“Finally, our study relies on a cross-sectional design, which does not allow for the identification of causal effects. While our models account for more potential confounding variables than earlier observational studies in this area of research, we cannot rule out the possibility of reverse-causality or a confounding influence of other factors that are not included in our models. Further investigation of the causal nature of the relationships between alcohol intake and brain anatomy (e.g., via longitudinal studies or natural experiments) would be of interest.”

Third, the observed brain deviations are not directly associated with a brain health problem. However reasonable it is to infer the existence of deficits or adverse effects as noted in the manuscript title, it is not clear from this study that there is a functional problem associated with the structural findings.

We agree that our findings do not provide direct evidence for a functional brain problem, and removed the term “brain health” from the manuscript.

The One sentence Summary now reads: *“Moderate alcohol intake, i.e., consuming one or more alcohol units daily, is associated with reduced gray matter and white matter volumes across the brain.”*

REVIEWER COMMENTS

Reviewer #5 (Remarks to the Author):

The authors have done an excellent job adjusting their presentation to take into account earlier criticisms. I have no additional concerns.